# CLEAR: An Information-Theoretic Framework for Distraction-Free Representation Learning in Visual Offline RL

## Abstract

Visual offline RL aims to learn an optimal policy for visual domains, solely from the pre-collected dataset comprised of actions taken on visual observations. Prior works on visual RL typically learn a dynamics model by extracting a latent state representation. However, the learned representation would contain factors irrelevant to control when there are distractions in the visual observations. These nuisance factors introduced by the distraction further exacerbates the difficulties of learning a good policy in the offline RL setting. In this work, we formalize the visual offline RL setting as a Partially Observable Markov Decision Process with exogenous variables (ExoPOMDP) and identify these problems with previous approaches under an information-theoretic lens. To overcome these challenges, we propose CLEAR (**C**ontrollable **L**atent State **E**xtr**A**cto**R**) for visual offline RL, which learns the dynamics model of a succinct agent-centric state representation that is consistent with the underlying ExoPOMDP. We empirically demonstrate that CLEAR is able to outperform baselines on the DeepMind Control Suite with various types of distractions and perform consistently well across these distractions. We further provide qualitative analysis on the results showing that our approach successfully disentangles the distraction factors from the agent-centric state representation.

## 1 Introduction

Offline Reinforcement Learning (Offline RL) (Lange et al., 2012; Levine et al., 2020) aims to learn policies solely from a fixed dataset of trajectories without any further access to the environment. While many offline RL algorithms have been proposed (Fujimoto et al., 2019; Kumar et al., 2020; Fujimoto & Gu, 2021), much of the recent progress has been limited to datasets which assume access to the underlying state of the environment (Fu et al., 2020). However, many datasets collected from real-world scenarios (e.g. autonomous driving (Yu et al., 2020) and robotics (Vuong et al., 2023)) consists of visual observations rather than state information. In these partially observable settings, extracting representations which capture the underlying state of the environment becomes critical to learn good policies.

However, inferring the ground-truth state from a sequence of image observations and learning a good offline RL policy is non-trivial. This is due to the fact that visual observations often contain complex distractions (e.g. background screens playing video advertisements or birds flying in the sky) which are irrelevant to the control task at hand (illustrated in Figure 1). The generalization challenges of offline RL (Fujimoto et al., 2019; Kumar et al., 2019) are further exacerbated by the presence of these distractions since they may spuriously correlate with the task. Thus, one of the keys to successful visual offline RL is to learn succinct agent-centric representations that capture the ground-truth state which are free from these distractions.

To address the challenge of partial observability in visual RL, one of the standard approaches is to learn the latent state dynamics model by maximizing the likelihood of the observed trajectory (Hafner et al., 2019; Lee et al., 2020; Hafner et al., 2020; Hwang et al., 2023). However, as we show through the experiments, we find that the learned representations still contain superfluous information irrelevant to control in the presence of distractions.

Figure 1: Visual observations consist of a controllable agent and distractions which are uncontrollable and unrelated to the task. Here we show samples from the (a) Cheetah-Run dataset with Video distractions and (b) Walker-Walk dataset with $2 \times 2$ Grid distractions that we will use in our main experiments.

In this work, we provide an information-theoretic framework for addressing this problem and learning distraction-free representations. We start by formalizing the visual RL problem as a Partially Observable Markov Decision Process with exogenous variables (ExoPOMDP). Under ExoPOMDP, we identify the main reasons why a latent state representation extracted by learning a single dynamics model, despite having an information bottleneck term, cannot be minimal when observations contain distractions. Specifically, previous approaches (Hafner et al., 2019; Lee et al., 2020) maximize the lower bound of an objective which maximizes predictive information while imposing the Markov property under an information-theoretic lens (Hwang et al., 2023). We show that in ExoPOMDPs, the learned representations may still contain superfluous information irrelevant for control. To overcome these shortcomings of previous approaches, we propose CLEAR (**C**ontrollable **L**atent State **E**xtr**A**cto**R**), which models both the agent-centric latent state dynamics as well as the distractions through separate encoders whose representations are disentangled. To train CLEAR, we introduce a regularized objective which additionally encourages the learned agent-centric state representation to be influenced or controlled by actions. Through this information-theoretic perspective, CLEAR provides a principled representation learning procedure that is consistent with the underlying ExoPOMDP.

Finally, we conduct experiments on a series of datasets with various degrees of distractions on the DeepMind Control Suite (Tassa et al., 2018), closely following the settings in (Lu et al., 2023; Islam et al., 2023). We show empirically that our method performs consistently well across these distractions and outperforms baselines especially for more dynamic distractions. We further provide qualitative results showing that our approach successfully disentangles the distraction variables from agent-centric ones.

## 2 BACKGROUND

### 2.1 EXOPOMDP FOR VISUAL OFFLINE REINFORCEMENT LEARNING

In this work, we attempt to explicitly model the distractions that exist in visual observations. The distractions can be characterized as a factor that 1) does not affect the reward function, 2) is unaffected by action, 3) is independent of the agent state, and 4) is present in the observation. Based on these characteristics, distractions then can be defined by exogenous random variables following prior work (Efroni et al., 2022).

More formally, we model the visual RL problem as a Partially Observable Markov Decision Process with exogenous variables (ExoPOMDP). An ExoPOMDP is defined by $\langle S, E, A, O, p^s, p^e, \mu_0^s, \mu_0^e, q, r, \gamma \rangle$ where $S$ is the set of latent ground-truth states $s$, $E$ is the set of latent exogenous factors $e$, $A$ is the set of actions $a$, $O$ is the set of observations $o$, $p^s(s_{t+1}|s_t, a_t)$ is the state transition distribution, $p^e(e_{t+1}|e_t)$ is the exogenous factor transition distribution, $\mu_0^s(s_0)$ is the initial state distribution, $\mu_0^e(e_0)$ is the initial distribution of the exogenous factor, $q(o_t|s_t, e_t)$ is the emission distribution, $r(s_t, a_t)$ is the reward function, and $\gamma$ is the discount factor. The graphical model of an ExoPOMDP is depicted in Figure 2. Importantly, the exogenous factors $e$ aim to satisfy the aforementioned 4 properties of distractions. Note that an ExoPOMDP does not make the block structure assumption used in Exogenous Block MDPs (EX-BMDPs) (Islam et al., 2023), where the state and exogenous components can be recovered from each observation without considering the dynamics.

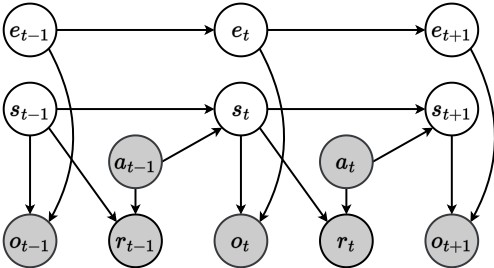

Figure 2: Graphical model of an ExoPOMDP where a POMDP is augmented with exogenous variables to model the distractions present in the observation despite not influencing the task.

The objective of reinforcement learning (RL) is to find a policy that maximizes the sum of discounted expected return $\pi^* = \operatorname{argmax}_\pi \mathbb{E}_\pi[\sum_{t=0}^\infty \gamma^t r(s_t, a_t)]$. In visual RL, the agent is only given observation $O$ instead of ground-truth states $S$. In visual offline RL, instead of access to the environment, the agent is given a dataset $D = \{(o_i, a_i, r_i, o_i')\}_{i=1}^N$ to find the optimal policy. Since the dataset is fixed, we take a two-step training approach where we pretrain to learn the representations from the fixed dataset and then train the (offline) RL agent on top of the frozen representations. This work focuses on the representation learning step, and evaluate the learned representations with an off-the-shelf offline RL method, TD3+BC (Fujimoto & Gu, 2021).

## 2.2 NEGATIVE EFFECTS OF SUPERFLUOUS INFORMATION IN EXOPOMDPS

Previous approaches (Hafner et al., 2019; Lee et al., 2020) aim to learn a latent state representation $\hat{s}_t$ using a stochastic encoder $p_\theta(\hat{s}_t|\hat{s}_{t-1}, a_{t-1}, o_t)$ which is parameterized by $\theta$. Here, we derive these works from an information-theoretic perspective and identify its shortcomings under the ExoPOMDP model. From the graphical model in Figure 2, we observe that the ground-truth state is predictive of future observation (i.e. $\langle S_{t-1}, A_{t-1}\rangle$ and $O_t$ are dependent) and Markovian (i.e. $\langle S_{t-1}, A_{t-1}\rangle$ and $O_t$ are conditionally independent given $S_t$). Thus, we wish our encoder to maximize the predictive information while enforcing the Markov property by maximizing the following objective function

$$J_{\text{State}}(\theta) \triangleq \underbrace{I_\theta(\hat{S}_{t-1}, A_{t-1}; O_t)}_{\text{predictive information}} - \underbrace{I_\theta(\hat{S}_{t-1}, A_{t-1}; O_t|\hat{S}_t)}_{\text{Markovian objective}}. \tag{1}$$

Since the mutual information (MI) terms are intractable, one may derive a variational lower-bound which results in an objective equivalent to the ELBO of an SSM (state-space model) commonly used in prior works (Hafner et al., 2019; Lee et al., 2020). This is observed in Hwang et al. (2023) and we provide details of this equivalence in Appendix B.

Upon maximizing equation 1, the Markovian objective on the second term can be minimized to 0 due to the non-negativity of conditional MI and thus induce a Markovian representation. For the predictive information, we can decompose it into two components that resemble the decomposition in supervised learning (Federici et al., 2020) as

$$\underbrace{I_{\theta^*}(\hat{S}_{t-1}, A_{t-1}; O_t)}_{\text{predictive information}} = \underbrace{I(S_{t-1}, A_{t-1}; S_t)}_{\text{state transition information}} + \underbrace{I_{\theta^*}(\hat{S}_{t-1}, A_{t-1}; O_t|S_t)}_{\text{superfluous information}}, \tag{2}$$

where $\theta^*$ denotes the optimal encoder parameter. While the representation contains information about the state transition dynamics, there is no mechanism to constrain the superfluous information on the right-hand side. Intuitively, this superfluous information corresponds to an exogenous factor since it characterizes the amount of information contained in the representation $\hat{S}_{t-1}$ about future observations $O_t$ even after observing future ground-truth state $S_t$. Thus, without any mechanism to constrain it, there is no guarantee that the learned representation will be minimal. Furthermore, careful readers might note that the superfluous information $I_\theta(\hat{S}_{t-1}, A_{t-1}; O_t|S_t)$ is dependent on $\theta$. However, it is conditioned on $S_t$ which is unobservable and thus cannot be computed nor minimized directly.

As demonstrated in the experimental results in Section 5, the presence of any superfluous information that spuriously correlates with the task may exacerbate the difficulties of learning a good

policy in offline RL. In Table 1, we show results for running TD3+BC (Fujimoto & Gu, 2021) on top of representations learned via SLAC (Lee et al., 2020), which serves as a representative method of $J_{\text{State}}$ and one of our baselines. The results are shown for three different environments from the DeepMind Control Suite (Tassa et al., 2018) with different degrees of distractions (Clean, Video, and $2 \times 2$ Grid) of increasing levels of difficulty. [1] Note that while SLAC performs well on the Clean setup, its performance consistently drops as distractions are introduced in the observation. Hence, it is evident that learning a representation that is free of superfluous information is crucial for learning good policies in offline RL.

## 3 METHOD

The analysis in Section 2 showed that nuisance factors in the form of exogenous variables may negatively affect performance in offline RL. In this section, we introduce CLEAR (Controllable Latent State Extractor), which learns succinct agent-centric representations that are robust to these nuisance factors.

### 3.1 LEARNING DISENTANGLED REPRESENTATIONS

In order to learn succinct agent-centric representations and exclude the superfluous information, our approach relies on learning two sets of representations ($\hat{S}_t$ and $\hat{E}_t$) which aim to capture both the state and exogenous variables ($S_t$ and $E_t$) independently. To learn the two sets of representations, we employ two stochastic encoders $p_\theta(\hat{s}_t|\hat{s}_{t-1}, a_{t-1}, o_t)$ and $p_\theta(\hat{e}_t|o_t)$ where we use $p_\theta$ to denote all stochastic encoder distributions parameterized by $\theta$.

We start by formulating the objective based on the desired properties of the ground-truth states and exogenous variables, following the analysis in Section 2.2. We observe that the ground-truth state is predictive and Markovian i.e. $\langle S_{t-1}, A_{t-1} \rangle$ and $O_t$ are dependent but conditionally independent given $S_t$ as mentioned in the previous section. Additionally, the state and exogenous variables are disentangled i.e. $S_t$ and $E_t$ are independent but conditionally dependent given $O_t$. Thus, to maximize predictive information while enforcing the Markov property and disentanglement on our learned representations, we wish to maximize the following objective function:

$$J(\theta) \triangleq J_{\text{State}}(\theta) + \underbrace{I_\theta(\hat{E}_t; \hat{S}_t|O_t) - I_\theta(\hat{E}_t; \hat{S}_t)}_{\text{disentanglement objective}}$$

$$= I_\theta(\hat{S}_t; O_t) - I_\theta(\hat{S}_t; O_t|\hat{S}_{t-1}, A_{t-1}) + I_\theta(\hat{E}_t; O_t|\hat{S}_t) - I_\theta(\hat{E}_t; O_t)$$

$$= I_\theta(\hat{S}_t, \hat{E}_t; O_t) - I_\theta(\hat{S}_t; O_t|\hat{S}_{t-1}, A_{t-1}) - I_\theta(\hat{E}_t; O_t). \tag{3}$$

We re-arrange the objective by employing the identity of interaction information in the second line and chain rule of MI in the third line.

All MI terms in equation 3 are intractable since each term involves the unknown data distribution $p_D$. However, we can derive a lower-bound by introducing variational distributions $q_\phi(o_t|\hat{s}_t, \hat{e}_t)$, $q_\phi(\hat{s}_t|\hat{s}_{t-1}, a_{t-1})$, and $q_\phi(\hat{e}_t)$ for the intractable $p_\theta(o_t|\hat{s}_t, \hat{e}_t)$, $p_\theta(\hat{s}_t|\hat{s}_{t-1}, a_{t-1})$, and $p_\theta(\hat{e}_t)$, respectively. Then, the lower-bound is given as

$$J(\theta) \geq \mathbb{E}_{p_D, p_\theta}\left[\log q_\phi(o_t|\hat{s}_t, \hat{e}_t)\right] + H(O_t) \tag{4}$$

$$- D_{KL}(p_\theta(\hat{s}_t|o_t, \hat{s}_{t-1}, a_{t-1})||q_\phi(\hat{s}_t|\hat{s}_{t-1}, a_{t-1})) - D_{KL}(p_\theta(\hat{e}_t|o_t, \hat{e}_{t-1})||q_\phi(\hat{e}_t))$$

$$\triangleq J_{\text{ELBO}}(t; \theta, \phi), \tag{5}$$

where $H(O_t)$ in equation 4 is determined by the fixed dataset and thus constant. Since the lower-bound resembles the combination between the ELBO of an SSM (Hafner et al., 2020; Lee et al., 2020) and VAEs (Kingma & Welling, 2014), we name this objective $J_{\text{ELBO}}$. Maximizing $J_{\text{ELBO}}$ with respect to the encoder parameter $\theta$ and variational distribution parameters $\phi$ will maximize the lower-bound of equation 3 and fit the variational distribution to their respective intractable distributions. We provide a full derivation for equation 5 in Appendix C.1.

However, since we have two latent variables and we optimize it via stochastic gradient descent with a fixed dataset, using equation 5 alone is prone to local optima. This local optima includes, for

---

[1]We provide details on the dataset and experimental setup in Section 5.

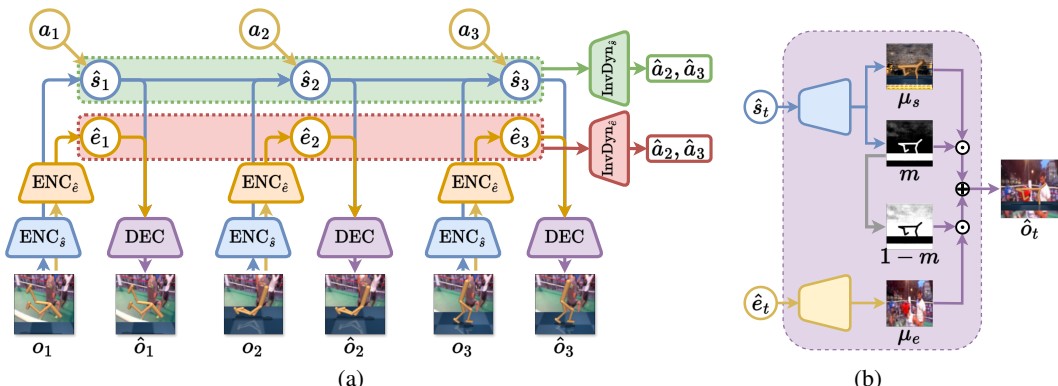

Figure 3: Overview of CLEAR. (a) Given a sequence of observations and actions, two sequences of representations are extracted via two sets of encoders $p_\theta(\hat{s}_t|\hat{s}_{t-1}, a_{t-1}, o_t)$ and $p_\theta(\hat{e}_t|o_t)$. Then, the two sets of representations are decoded to reconstruct the observations and do inverse dynamics prediction. (b) The decoder $q_\phi(o_t|\hat{s}_t, \hat{e}_t)$ which reconstructs observations has a compositional structure.

instance, "flipped" representations where $\hat{S}_t$ captures $E_t$ while $\hat{E}_t$ captures $S_t$. We provide empirical evidence for this in our ablation results in Section 5.3. In the next section, we will add additional regularization terms to alleviate these issues.

## 3.2 REGULARIZATION FOR ACTION CONTROLLABILITY

One key distinguishing characteristic between $S$ and $E$ lies in their dependency on actions; $S$ is influenced by or responsive to actions while $E$ is not. Since the transition is induced by the action, $a_t$ should be inferable given $s_t$ and $s_{t+1}$. Thus, we encourage any two consecutive states to be informative of the in-between action by maximizing

$$I_\theta(A_t; \hat{S}_t, \hat{S}_{t+1}) \geq \mathbb{E}_{p_D, p_\theta}[\log q_\phi(a_t|\hat{s}_t, \hat{s}_{t+1})] + H(A_t)$$
$$\triangleq J_{\text{InvDyn-S}}(t; \theta, \phi), \tag{6}$$

where it can be lower-bounded by introducing additional variational distributions $q_\phi(a_t|\hat{s}_t, \hat{s}_{t+1})$ to approximate the intractable $p_\theta(a_t|\hat{s}_t, \hat{s}_{t+1})$. This is equivalent to inverse dynamics prediction.

Conversely, for $\hat{E}$, we wish to minimize $I_\theta(A_t; \hat{E}_t, \hat{E}_{t+1})$ since all the information necessary to predict the action should be in the $\hat{S}$. However, deriving the upper-bound of $I_\theta(A_t; \hat{E}_t, \hat{E}_{t+1})$ is non-trivial. Instead, we derive a lower-bound and employ a min-max optimization procedure as follows

$$\min_\theta I_\theta(A_t; \hat{E}_t, \hat{E}_{t+1}) \approx \min_\theta \max_\psi \mathbb{E}_{p_D, p_\theta}[\log q_\psi(a_t|\hat{e}_t, \hat{e}_{t+1})] + H(A_t)$$
$$= \max_\theta \min_\psi \mathbb{E}_{p_D, p_\theta}[-\log q_\psi(a_t|\hat{e}_t, \hat{e}_{t+1})] - H(A_t)$$
$$\triangleq \max_\theta \min_\psi -J_{\text{InvDyn-E}}(t; \theta, \psi), \tag{7}$$

where we use $\psi$ as the variational distribution parameter. We can optimize equation 7 in an alternating fashion by updating $\psi$ to do inverse dynamics prediction and then updating $\theta$ to make the inverse dynamics prediction worse.

## 3.3 CLEAR: CONTROLLABLE LATENT STATE EXTRACTOR

In summary, our regularized optimization objective is

$$\max_{\theta, \phi} \min_\psi J_{\text{ELBO}}(t; \theta, \phi) + J_{\text{InvDyn-S}}(t; \theta, \phi) - J_{\text{InvDyn-E}}(t; \theta, \psi). \tag{8}$$

Figure 3a illustrates our overall method. Our proposed objective is general in the sense that we can employ other types of bounds such as the contrastive loss as opposed to the variational bound

(image reconstruction and inverse dynamics prediction). We opt to use the simple variational bound since in practice we found it to work well, in line with the observation in (Hafner et al., 2020). Once the model is trained, the encoder $p_\theta(\hat{s}_t|\hat{s}_{t-1}, a_{t-1}, o_t)$ will be frozen and utilized to extract representations for the downstream offline RL task.

For practical purposes, we use two different constants for the two KL terms in $J_{\text{ELBO}}$, which correspond to the information bottleneck terms. We found that doing so controls the amount of information that passes through each encoder and improves the performance.

Lastly, assuming the state variables and exogenous variables occupy different parts of the visual observation, we employ a compositional decoder commonly used in object-centric representation learning (Greff et al., 2019; Locatello et al., 2020). We parameterize $q_\phi(o_t|\hat{s}_t, \hat{e}_t)$ to be Gaussian with a learnable mean $\mu$ and a fixed standard deviation where we model each pixel independently. Inferred state variables $\hat{s}_t$ and exogenous variables $\hat{e}_t$ are decoded separately. Then, for each pixel, $\hat{s}_t$ is decoded to output the pixel mean $\mu_s$ and a mask $m \in [0, 1]$ while $\hat{e}_t$ is decoded to output the pixel mean $\mu_e$. The two pixel means are then combined using the mask as $\mu = m\mu_s + (1 - m)\mu_e$. Figure 3b illustrates our compositional decoder.

## 4 RELATED WORK

**Latent Dynamics Models**  To address the challenge of partial observability in visual RL, prior works learn latent variable models by maximizing the lower-bound of the log-likelihood of the observed trajectory that recovers the latent state dynamics (Hafner et al., 2019; Lee et al., 2020; Hafner et al., 2020; Hwang et al., 2023). The learned model then can be applied to extract representation for model-free RL (Lee et al., 2020; Hwang et al., 2023), planning (Hafner et al., 2019), and model-based RL (Hafner et al., 2020). However, as we have seen in Section 2.2, the representations extracted through this approach can include superfluous information which is problematic for learning good policies in offline RL.

**Task-Relevant Representations**  To eliminate the distraction (superfluous information) from the representation, prior works have incorporated the concept of task-relevance. DRIBO (Fan & Li, 2022) extracts task-relevant representation via the multi-view information bottleneck (Federici et al., 2020), obtaining two views by data augmentation. However, generating two views by data augmentation does not guarantee the mutual redundancy assumption necessary for their method. TiA (Fu et al., 2021) takes a similar approach to ours and attempts to learn two sets of representations. However, TiA uses two identical encoders and uses reward to differentiate between task-relevant and task-irrelevant feature. Using reward to identify task-relevance is problematic since rewards may be sparse and/or dependent only on the subset of the agent state. Denoised MDP (Wang et al., 2022) goes one step further by modeling three sets of representations and categorising features based on its task-relevance and controllability. However, it still only uses reward and makes the problem underdetermined since they rely solely on the reward to separate three sets of representations. RePo (Zhu et al., 2023) avoids observation reconstruction by only predicting the reward to obtain task-relevant representations, and thus inherits the same problems faced by TIA in utilizing reward. This issue has been studied in detail in the ablation studies in (Hafner et al., 2020).

**Control-Relevant Representations**  Finally, another line of work uses the notion of control-relevance to remove the distractions. The single-step inverse dynamics (predicting action at time $t$ given observations at time $t$ and $t + 1$) has been empirically observed to be effective to learn a representation for control (Agrawal et al., 2016; Pan et al., 2022; Brandfonbrener et al., 2023; Paster et al., 2021). Intuitively, the representation only needs to capture features that are necessary to predict the action given the transition. However, Rakelly et al. (2021) showed that the representations learned via inverse dynamics is not sufficient for control. To resolve this issue, the multi-step inverse dynamics (predicting action at time $t$ given observations at time $t$ and $t+k$ where $k$ is a hyperparameter) has been proposed (Efroni et al., 2022; Lamb et al., 2023). ACRO (Islam et al., 2023) utilizes the multi-step inverse dynamics in the context of offline RL. InfoGating (Tomar et al., 2023) extends ACRO by learning a sparse mask to mask out the irrelevant part of visual observation. However, the multi-step inverse dynamics does not fully resolve the problem (Levine et al., 2024) and is an inherently ill-posed problem since there are multiple actions that can achieve the same transition. Unlike previous approaches, we derive the single-step inverse dynamics from an information-theoretic per-

spective and use it as a regularization term instead of the main objective. InfoPower (Bharadhwaj et al., 2022) learns a latent state dynamics and avoids reconstruction by using contrastive loss while regularizing the model with inverse dynamics prediction. However, using contrastive loss tends to perform poorly in practice when compared to the reconstruction loss (Hafner et al., 2020). Similar to our method, Iso-Dream (Pan et al., 2022) utilizes an additional encoder and regularizes its model using inverse dynamics. However, their approach uses three latent variables with one regularization term. This makes the model underspecified and prone to local optimas which can negatively affect offline RL performance as we show in the experiments, similar to Denoised-MDPs (Wang et al., 2022). On the other hand, we derive our method in a principled manner using mutual information to reflect the underlying ExoPOMDP resulting in a more general method in the sense that ours is simpler and we can employ other types of bounds.

## 5 Experiments

**Datasets** To validate the effectiveness of CLEAR against various levels of distractions, we evaluate our algorithm on the DeepMind Control Suite (Tassa et al., 2018), which is a standard benchmark in visual offline RL (Lu et al., 2023; Islam et al., 2023). Since v-d4rl (Lu et al., 2023) only provides datasets for image observations with static backgrounds, we construct our own set of datasets which also includes dynamic distractions.

For each dataset, we generate four levels of varying difficulties of distractions by adjusting the types of distractions present in the observation. The `easy` level has a static background which is used in the original observations (**Clean**). For the `medium` level, we introduce correlated distractions by using, as the background, a single video which repeats for every episode (**SV**) and four videos which change every episode (**MV**). Lastly, for the `hard` level, we make a $2 \times 2$ grid where we put the agent that we can control on the top-left of the grid. For the rest of the grid, we put similar agents which are controlled by a random uniform policy ($\mathbf{2 \times 2}$). Figure 1 shows a sample of the Cheetah-Run dataset with Video distraction and Walker-Walk dataset with $2 \times 2$ Grid distractions. We provide more details about the dataset construction as well as some samples of the dataset in Appendix G. [2]

We evaluate on three sets of environments, namely Hopper-Hop, Walker-Walk, and Cheetah-Run. In order to ensure a fair comparison, we collect medium-expert datasets which have been shown to be an appropriate level for the baselines to perform well (Lu et al., 2023).

**Baselines** Following the prior work (Lu et al., 2023), we use TD3+BC (Fujimoto & Gu, 2021) as the offline RL algorithm to evaluate the learned representations for all baselines as well as for CLEAR. We include **SLAC** (Lee et al., 2020), **TiA** (Fu et al., 2021), InfoPower (Bharadhwaj et al., 2022), **Iso-Dream** (Pan et al., 2022), Denoised MDP (**Den-MDP**) (Wang et al., 2022), and **RePo** (Zhu et al., 2023) as baselines which learn the latent state dynamics. Although TiA, Iso-Dream, Den-MDP, and RePo were originally proposed as model-based methods, we can use the variational posterior to extract representations similar to what was done in (Wang et al., 2022). Additionally, we include **DrQ-v2** (Yarats et al., 2022), **ACRO** (Islam et al., 2023), and **InfoGating** (Tomar et al., 2023). These additional methods do not learn the latent state dynamics but instead take a stack of consecutive frames as a state. Lastly, we train TD3+BC using the ground-truth state as an upper-bound on the performance to normalize the score. A score of 100 means that it performs as good as using the ground-truth state. We provide further implementation details and hyperparameters in Appendix H.

### 5.1 Offline RL Results

Table 1 shows the main results of our experiments. [3] We run each experiment over 5 random seeds and report the average normalized score and its standard error. Since the underlying dataset quality is the same, the desired result is for the score to be invariant across different distractions.

First, we observe that in all environments and distraction levels, CLEAR significantly outperforms SLAC, TiA, Den-MDP, and RePo, which all learn latent dynamics models. Again, this result pro-

---

[2]Our anonymous code is available at `https://anonymous.4open.science/r/anonymous-clear-EFFC`

[3]We also provide the results for two additional baselines (Single-Step Inverse Dynamics and DINO-v2 (Oquab et al., 2024)) in Appendix D.

|  |  | SLAC | TiA | InfoPower | Den-MDP | Iso-Dream | RePo | DrQ-v2 | ACRO | InfoGating | CLEAR |
|---|---|---|---|---|---|---|---|---|---|---|---|
| Hopper | Clean (easy) | 88.2 ± 3.6 | 20.0 ± 2.8 | 1.2 ± 0.5 | 25.9 ± 2.7 | 69.8 ± 7.5 | 4.5 ± 1.2 | 88.5 ± 2.6 | 73.7 ± 3.9 | 78.1 ± 2.8 | **104.9 ± 2.8** |
|  | SV (medium) | 14.6 ± 1.2 | 1.9 ± 0.6 | 1.0 ± 0.2 | 22.7 ± 3.0 | 28.0 ± 5.0 | 5.0 ± 1.0 | 64.1 ± 2.4 | 64.0 ± 2.4 | **82.9 ± 2.1** | 60.5 ± 5.1 |
|  | MV (medium) | 4.6 ± 0.9 | 2.3 ± 0.3 | 1.5 ± 0.0 | 10.0 ± 3.3 | 22.2 ± 7.2 | 3.9 ± 0.3 | 49.7 ± 2.0 | 51.7 ± 1.5 | **62.0 ± 4.0** | 39.8 ± 11.5 |
|  | 2 × 2 (hard) | 5.4 ± 0.9 | 0.1 ± 0.1 | 0.9 ± 0.4 | 8.3 ± 1.2 | 25.5 ± 4.6 | 3.6 ± 0.8 | 27.0 ± 3.9 | 35.1 ± 3.1 | 44.7 ± 4.1 | **50.5 ± 4.2** |
| Walker | Clean (easy) | 74.5 ± 11.6 | 79.6 ± 2.6 | 3.8 ± 0.1 | 38.5 ± 3.2 | **83.5 ± 8.5** | 38.8 ± 3.2 | 75.6 ± 2.6 | 89.7 ± 1.7 | 89.0 ± 1.1 | 89.9 ± 2.0 |
|  | SV (medium) | 79.9 ± 3.6 | 80.1 ± 2.7 | 2.8 ± 0.0 | 50.9 ± 4.5 | **92.0 ± 1.3** | 35.8 ± 1.8 | 56.3 ± 1.7 | 88.3 ± 1.0 | 90.7 ± 1.4 | 87.6 ± 3.8 |
|  | MV (medium) | 68.1 ± 1.8 | 62.8 ± 4.5 | 2.8 ± 0.1 | 46.9 ± 2.3 | 84.3 ± 3.1 | 27.1 ± 5.9 | 62.3 ± 1.2 | **88.8 ± 1.9** | 83.4 ± 3.5 | 88.4 ± 1.8 |
|  | 2 × 2 (hard) | 44.5 ± 3.8 | 26.5 ± 3.2 | 2.1 ± 0.9 | 29.8 ± 3.0 | 80.1 ± 5.0 | 34.9 ± 3.3 | 45.7 ± 1.3 | 76.4 ± 2.0 | 81.3 ± 2.5 | **88.8 ± 2.4** |
| Cheetah | Clean (easy) | 95.0 ± 1.7 | 67.7 ± 6.2 | 24.1 ± 1.6 | 43.6 ± 3.9 | 56.5 ± 12.7 | 38.1 ± 5.7 | 85.3 ± 3.2 | 85.0 ± 3.1 | 72.5 ± 3.0 | **96.5 ± 0.6** |
|  | SV (medium) | 72.6 ± 4.2 | 58.9 ± 5.7 | 24.8 ± 2.7 | 64.6 ± 3.9 | **94.5 ± 1.2** | 37.1 ± 4.4 | 73.6 ± 1.0 | 79.9 ± 0.8 | 86.7 ± 1.8 | 96.7 ± 1.5 |
|  | MV (medium) | 54.7 ± 4.0 | 36.0 ± 3.7 | 25.4 ± 2.1 | 45.5 ± 2.4 | **94.0 ± 3.1** | 43.1 ± 3.8 | 60.8 ± 2.5 | 59.1 ± 3.5 | 68.1 ± 4.5 | 95.8 ± 1.1 |
|  | 2 × 2 (hard) | 46.2 ± 4.7 | 29.9 ± 1.3 | 20.7 ± 0.1 | 39.0 ± 2.2 | 32.1 ± 3.0 | 37.7 ± 1.0 | 51.0 ± 2.7 | 43.2 ± 1.6 | 47.4 ± 5.0 | **79.1 ± 4.2** |

Table 1: Average normalized score and its standard error over 5 seeds on the DeepMind Control Suite for Clean, Single Video (SV), Multiple Videos (MV) and 2 × 2 Grid distractions.

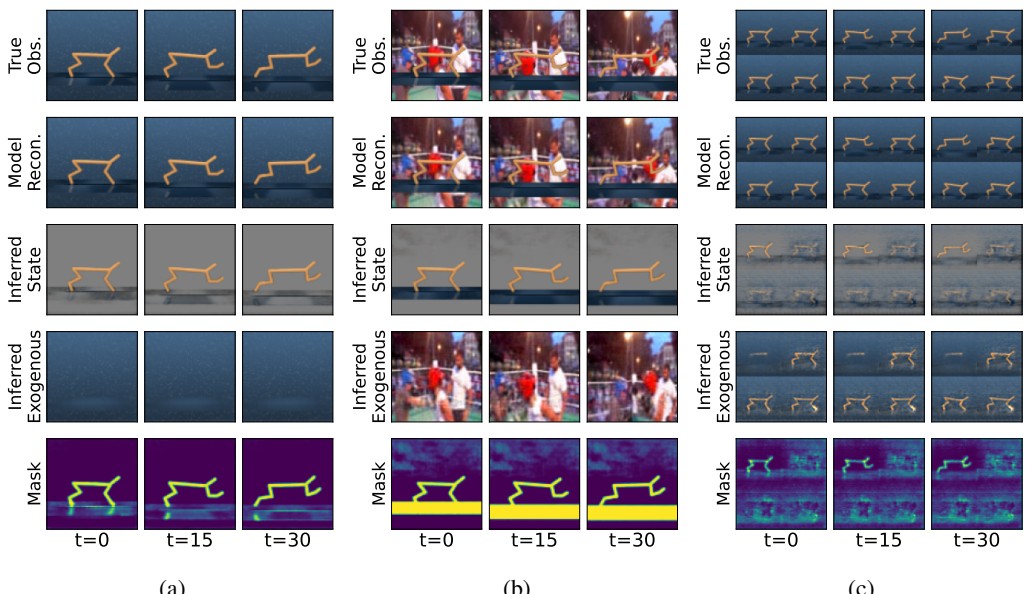

Figure 4: Reconstruction results of CLEAR for Cheetah-Run dataset on the distraction level of (a) Clean, (b) Multiple Videos, and (c) 2 × 2 Grid. Starting from first row to the last, the figure shows the original image observations, the reconstructed observations, the inferred state, the inferred exogenous component, and the mask used to combine the inferred state and exogenous component.

vides evidence for the negative implications of having superfluous information in the learned representations, as we have discussed in Section 2.2. Iso-Dream has comparable performance in Walker and Cheetah environment with video distractions but not in 2 × 2 Grid distraction which is likely due to its underspecified model as we mentioned in Section 4. Nonetheless, it suggests that control-relevant representation which uses inverse dynamics prediction is more effective than task-relevant representation which uses reward prediction as regularization. It is evident that CLEAR is the only latent dynamics method that can consistently remove superfluous information and maintain a level of invariance in the offline RL performance.

For Hopper, while CLEAR is unable to achieve the desired distraction-robust performance, it still performs on par with the strongest baselines, namely InfoGating. The improvement in performance from SLAC to CLEAR suggests that CLEAR is still able to significantly remove superfluous information from its representations albeit not entirely.

For Walker, Iso-Dream, ACRO, InfoGating, and CLEAR are able to achieve distraction-robust performance in both Single Video and Multiple Videos distractions. However, baselines fail at the 2 × 2 Grid distraction, suggesting it struggles to identify which Walker among the four is controllable. CLEAR's performance, on the other hand, is equal to that of the Clean dataset with no distractions.

| | | SLAC | TiA | InfoPower | Den-MDP | Iso-Dream | RePo | ACRO | InfoGating | CLEAR |
|---|---|---|---|---|---|---|---|---|---|---|
| Hopper | Clean (easy) | **0.92 ± 0.04** | 1.09 ± 0.05 | 2.17 ± 0.06 | 1.30 ± 0.05 | **0.97 ± 0.09** | 1.92 ± 0.08 | 1.08 ± 0.02 | 1.08 ± 0.03 | **1.04 ± 0.15** |
| | SV (medium) | 1.86 ± 0.07 | 2.79 ± 1.09 | 2.23 ± 0.05 | **1.17 ± 0.06** | **1.35 ± 0.15** | 2.03 ± 0.05 | 1.26 ± 0.04 | **1.18 ± 0.03** | **1.11 ± 0.10** |
| | MV (medium) | 2.94 ± 0.09 | 3.23 ± 1.39 | 2.16 ± 0.05 | **1.93 ± 0.90** | **1.81 ± 0.71** | 2.08 ± 0.03 | **1.43 ± 0.04** | 1.41 ± 0.06 | **1.58 ± 0.40** |
| | 2 × 2 (hard) | **1.26 ± 0.06** | 2.37 ± 0.11 | 2.23 ± 0.07 | 1.69 ± 0.10 | 1.36 ± 0.12 | 2.04 ± 0.10 | 1.55 ± 0.06 | 1.54 ± 0.06 | **1.15 ± 0.09** |
| Walker | Clean (easy) | **2.59 ± 0.04** | 2.92 ± 0.05 | 3.51 ± 0.15 | 2.79 ± 0.04 | **2.72 ± 0.19** | 4.17 ± 0.07 | 3.49 ± 0.03 | 3.38 ± 0.08 | **2.62 ± 0.06** |
| | SV (medium) | 3.55 ± 0.18 | 4.01 ± 0.24 | 3.61 ± 0.02 | 2.89 ± 0.07 | **2.52 ± 0.03** | 4.21 ± 0.06 | 3.69 ± 0.06 | 3.60 ± 0.05 | 2.99 ± 0.25 |
| | MV (medium) | 3.91 ± 0.22 | 4.19 ± 0.24 | 3.59 ± 0.12 | **2.87 ± 0.09** | **2.76 ± 0.19** | 4.20 ± 0.03 | 3.86 ± 0.08 | 3.70 ± 0.06 | **3.04 ± 0.32** |
| | 2 × 2 (hard) | 4.45 ± 0.09 | 5.92 ± 0.12 | 3.64 ± 0.14 | 3.93 ± 0.13 | 4.15 ± 0.33 | 4.30 ± 0.12 | 4.33 ± 0.07 | 4.23 ± 0.14 | **3.27 ± 0.07** |
| Cheetah | Clean (easy) | **0.83 ± 0.02** | 1.21 ± 0.06 | 2.61 ± 0.12 | 1.62 ± 0.04 | **0.85 ± 0.01** | 2.52 ± 0.05 | 1.81 ± 0.04 | 1.83 ± 0.05 | **0.88 ± 0.03** |
| | SV (medium) | 3.08 ± 0.11 | 4.08 ± 1.11 | 2.57 ± 0.04 | 2.97 ± 0.20 | **1.51 ± 0.26** | 2.70 ± 0.11 | 2.44 ± 0.05 | 2.37 ± 0.04 | **1.22 ± 0.56** |
| | MV (medium) | 4.07 ± 0.06 | 5.34 ± 0.47 | 2.64 ± 0.13 | 3.00 ± 0.29 | **1.29 ± 0.16** | 2.58 ± 0.07 | 2.81 ± 0.07 | 2.78 ± 0.06 | **1.19 ± 0.22** |
| | 2 × 2 (hard) | 1.29 ± 0.02 | 1.76 ± 0.01 | 2.52 ± 0.09 | 1.60 ± 0.06 | 1.27 ± 0.02 | 2.75 ± 0.06 | 2.37 ± 0.03 | 2.37 ± 0.12 | **1.14 ± 0.04** |

Table 2: Average MSE and its standard deviation over 5 seeds on the ground-truth state regression task using linear model.

Lastly for Cheetah, CLEAR outperforms all baselines at all distraction levels, except Iso-Dream, which performs on-par with CLEAR on the video distractions. We note that the slight decrease in the $2 \times 2$ Grid distraction performance can be explained by the difficulty of distinguishing controllable and random uniform Cheetah agents as shown in the dataset sample in Figure 8 in the Appendix.

Figure 4 shows the qualitative results of the learned representations of CLEAR. We observe that our model is able to successfully disentangle the agent from the distractions in the observations. In the Clean and Multiple Videos dataset, the inferred state successfully removes all the background information from the representation. Interestingly, in the Multiple Videos dataset, the exogenous part can infer the occluded segment of the background. Lastly, in the hardest level of $2 \times 2$ Grid, the model is able to identify that the agent in the top-left corner is the one that is controllable.

Finally, we also show that CLEAR is able to generalize to unseen background distractions and outperforms the strongest baselines (See Appendix E for details).

## 5.2 GROUND-TRUTH STATE REGRESSION

To show how informative the learned state representation is about the ground-truth state, we perform linear regression to predict the ground-truth state using the pretrained frozen encoder. In addition to the original 400k timestep dataset for training, we collect an additional 100k timesteps as the validation set. Table 2 shows the average mean squared error (MSE) of predicting the ground-truth state on the validation set.

The result is consistent with our hypothesis. While SLAC predicts the ground-truth state well on the Clean setup, its prediction gets worse as distractions are introduced in the observation hinting the negative effect of superfluous information. CLEAR reliably has low MSE across environments and distractions. Additionally, the case where SLAC, Iso-Dream, ACRO, and InfoGating have high average normalized score in Table 1 translates to its representation having low MSE hinting the representation has high information regarding the ground-truth state. However, the reverse is not true (i.e. low MSE does not necessarily translate to good offline RL performance) as can be seen in SLAC $2 \times 2$ Grid in Hopper and Cheetah, Den-MDP in Hopper and Walker, and Iso-Dream in Hopper. Thus, it further supports our claim that superfluous information (i.e. information about distractions) makes learning good policies in offline RL more difficult.

## 5.3 ABLATION STUDY

Our method consists of one main objective $J_{\text{ELBO}}$ and one regularization term $J_{\text{InvDyn-S}} - J_{\text{InvDyn-E}}$. We perform ablations to see the importance of each term using Cheetah (Multiple Videos).

The average normalized score over 5 seeds is reported in Table 3. We first observe that solely performing inverse dynamics prediction results in poor representations which degrade the offline RL performance. This corroborates the analysis in prior works which found that inverse dynamics prediction results in overly aliased state representations (Rakelly et al., 2021; Islam et al., 2023). Furthermore, we find that the inverse dynamics regularization term helps stabilize the training procedure and improve overall performance.

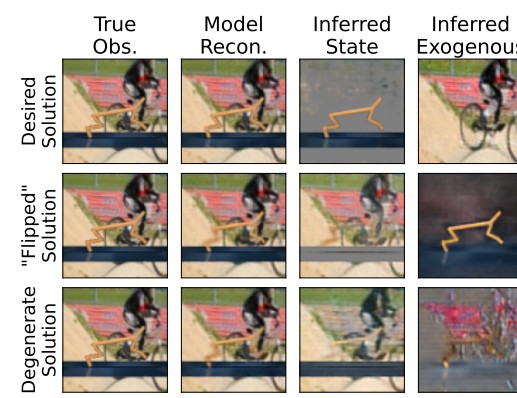

|  | $J_{\text{ELBO}}$ | $J_{\text{InvDyn}}$ | CLEAR |
|---|---|---|---|
| Clean (`easy`) | 37.8 ± 3.0 | 23.3 ± 0.7 | **96.5 ± 0.6** |
| SV (`medium`) | 60.3 ± 13.5 | 24.0 ± 0.3 | **96.7 ± 1.5** |
| MV (`medium`) | 62.0 ± 12.5 | 21.4 ± 1.2 | **95.8 ± 1.1** |
| 2 × 2 (`hard`) | 43.0 ± 4.8 | 24.5 ± 0.9 | **79.1 ± 4.2** |

Table 3: Ablation results on the Cheetah environment. $J_{\text{ELBO}}$ optimizes only equation 5 while $J_{\text{InvDyn}}$ optimizes only equation 6 and equation 7. Reported is the average and the standard error for 5 random seeds.

Figure 5: Qualitative resuls for $J_{\text{ELBO}}$ without regularization on three different random seeds.

Interestingly, we observe that the poor performance when maximizing $J_{\text{ELBO}}$ without any regularization is a result of different seeds converging to different representations. Figure 5 shows the differences qualitatively. Despite being able to reconstruct the original observation quite well, the information contained in the inferred state is different. The first row shows the desired solution which successfully captures the agent in the state representation while the video background is captured in the exogenous representation. In the second row, the solution is flipped and the state representation captures the video background while the exogenous representation captures the agent. This corresponds to the local optima discussed in the end of Section 3.1. Finally, the third row shows a degenerate solution where the disentanglement is unclear. These differences lead to some random seeds performing very poorly in downstream offline RL tasks, achieving normalized scores of 95.5, 38.2, and 59.6 for the desired, flipped, and degenerate solutions, respectively.

## 6 CONCLUSION

In this work, we presented CLEAR, which takes an information-theoretic approach to learning succinct agent-centric representations for visual offline RL. We introduced ExoPOMDPs and identified the shortcomings of previous approaches which learn latent state dynamics, namely the existence of superfluous information in the learned representations. CLEAR mitigates these issues through a separate encoder for learning the agent-centric and exogenous representations, trained by a regularized objective derived from the graphical model of the ExoPOMDP. We quantitatively and qualitatively validated our approach on the DeepMind Control Suite with varying levels of distractions. CLEAR outperformed previous baselines and demonstrated its ability to disentangle the agent-centric representations from the distraction factors, even with dynamic distractions.

## 7 REPRODUCIBILITY STATEMENT

We provide open access to the data and code (see Section 5 for the link). We provide details on the experimental setup (training details, dataset details, hyperparameters for both our method and baselines) in detail in the Appendices G and H.

## 8 ETHICS STATEMENT

Our work is primarily focused on extracting agent-centric representations which are invariant to various types of background distractions. Our research can be useful in many real-world control settings such as robotics and self-driving cars, where datasets containing image observations are available. However, safety issues such as crashes can arise when the representation fails to accurately capture the latent state, especially in complex real-world scenarios with dynamic and/or novel distractions.

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

## A CHARACTERIZING PREDICTIVE INFORMATION

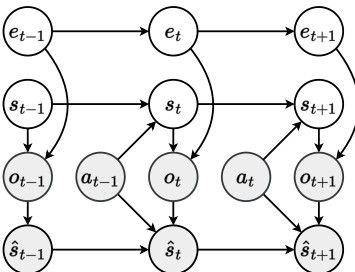

Figure 6: An augmented graphical model of the ExoPOMDP from Figure 2 where we augment it with observable $\hat{S}$ which is provided by our encoder $p_\theta(\hat{s}_t|\hat{s}_{t-1}, a_{t-1}, o_t)$. We do not visualize the observable reward $r(s_t, a_t)$ for visualization clarity.

In this section, we provide derivation for the equality provided in equation 2. For clarity, we provide an augmented version of Figure 2 where we augment it with an observable variable provided by our encoder $p_\theta(\hat{s}_t|\hat{s}_{t-1}, a_{t-1}, o_t)$ in Figure 6. Additionally, we rewrite $J_{\text{State}}$ below

$$J_{\text{State}}(\theta) \triangleq I_\theta(\hat{S}_{t-1}, A_{t-1}; O_t) - I_\theta(\hat{S}_{t-1}, A_{t-1}; O_t|\hat{S}_t).$$

We can derive an upper-bound of $J_{\text{State}}$ in ExoPOMDP as

$$J_{\text{State}}(\theta) \leq I_\theta(\hat{S}_{t-1}, A_{t-1}; O_t) \leq I(S_{t-1}, A_{t-1}; S_t) + I_\theta(\hat{S}_{t-1}, A_{t-1}; O_t|S_t).$$

The first part of inequality is achieved due to non-negativity of conditional mutual information i.e. $I_\theta(\hat{S}_{t-1}, A_{t-1}; O_t|\hat{S}_t) \geq 0$. For the second part of inequality, we break down the inequality into two parts: 1) $I_\theta(\hat{S}_{t-1}, A_{t-1}; S_t) \leq I(S_{t-1}, A_{t-1}; S_t)$ and 2) $I_\theta(\hat{S}_{t-1}, A_{t-1}; O_t) \leq I_\theta(\hat{S}_{t-1}, A_{t-1}; S_t) + I_\theta(\hat{S}_{t-1}, A_{t-1}; O_t|S_t)$.

1. $I_\theta(\hat{S}_{t-1}, A_{t-1}; S_t) \leq I(S_{t-1}, A_{t-1}; S_t)$

$$I_\theta(S_t; \hat{S}_{t-1}, A_{t-1}, S_{t-1}) = I_\theta(S_t; \hat{S}_{t-1}, A_{t-1}, S_{t-1})$$

$$I(S_t; S_{t-1}, A_{t-1}) + \underline{I_\theta(S_t; \hat{S}_{t-1}|S_{t-1}, A_{t-1})} = I_\theta(S_t; \hat{S}_{t-1}, A_{t-1}) + I_\theta(S_t; S_{t-1}|\hat{S}_{t-1}, A_{t-1})$$

$$I(S_t; S_{t-1}, A_{t-1}) \geq I_\theta(S_t; \hat{S}_{t-1}, A_{t-1})$$

2. $I_\theta(\hat{S}_{t-1}, A_{t-1}; O_t) \leq I_\theta(\hat{S}_{t-1}, A_{t-1}; S_t) + I_\theta(\hat{S}_{t-1}, A_{t-1}; O_t|S_t)$

$$I_\theta(\hat{S}_{t-1}, A_{t-1}; S_t, O_t) = I_\theta(\hat{S}_{t-1}, A_{t-1}; S_t, O_t)$$

$$I_\theta(\hat{S}_{t-1}, A_{t-1}; S_t) + I_\theta(\hat{S}_{t-1}, A_{t-1}; O_t|S_t) = I_\theta(\hat{S}_{t-1}, A_{t-1}; O_t) + I_\theta(\hat{S}_{t-1}, A_{t-1}; S_t|O_t)$$

$$I_\theta(\hat{S}_{t-1}, A_{t-1}; S_t) + I_\theta(\hat{S}_{t-1}, A_{t-1}; O_t|S_t) \geq I_\theta(\hat{S}_{t-1}, A_{t-1}; O_t)$$

Thus, upon maximization, we have the following equalities

1. $I_{\theta^*}(\hat{S}_{t-1}, A_{t-1}; O_t|\hat{S}_t) = 0$

2. $I_{\theta^*}(\hat{S}_{t-1}, A_{t-1}; O_t) = I(S_{t-1}, A_{t-1}; S_t) + I_{\theta^*}(\hat{S}_{t-1}, A_{t-1}; O_t|S_t)$

where $\theta^*$ denotes the optimal encoder parameter.

Note that if there are no exogenous variables, then $\langle \hat{S}_{t-1}, A_{t-1} \rangle \perp\!\!\!\perp O_t|S_t$ which means there is no superfluous information $I_\theta(\hat{S}_{t-1}, A_{t-1}; O_t|S_t) = 0$. This explains why SLAC (Lee et al., 2020) works well in the Clean dataset as demonstrated in Section 2.2.

## B  LOWER-BOUND OF $J_{\text{STATE}}$ AS ELBO OF AN SSM

In this section, we will derive the variational lower-bound of $J_{\text{State}}$ and show that it is equivalent to the ELBO of an SSM (Hafner et al., 2019; Lee et al., 2020).

$$J_{\text{State}}(\theta) \triangleq I_\theta(\hat{S}_{t-1}, A_{t-1}; O_t) - I_\theta(\hat{S}_{t-1}, A_{t-1}; O_t | \hat{S}_t) = I_\theta(\hat{S}_t; O_t) - I_\theta(\hat{S}_t; O_t | \hat{S}_{t-1}, A_{t-1})$$

$$\geq \mathbb{E}_{p_D, p_\theta} \big[ \log q_\phi(o_t | \hat{s}_t) \big] + H(O_t) - D_{KL}(p_\theta(\hat{s}_t | o_t, \hat{s}_{t-1}, a_{t-1}) || q_\phi(\hat{s}_t | \hat{s}_{t-1}, a_{t-1}))$$

In the first line, we rewrite $J_{\text{State}}$ and use the identity of interaction information. Lastly, the lower-bound is derived using similar techniques from Eq equation 10 and Eq equation 11.

Ignoring the constant $H(O_t)$ and extending the bound to a sequence of length $T$, we have

$$\mathbb{E}_{p_\theta(o_{\leq T}, \hat{s}_{\leq T}, a_{<T})} \Big[ \sum_{t=1}^{T} \log q_\phi(o_t | \hat{s}_t) - D_{KL}(p_\theta(\hat{s}_t | o_t, \hat{s}_{t-1}, a_{t-1}) || q_\phi(\hat{s}_t | \hat{s}_{t-1}, a_{t-1})) \Big] \tag{9}$$

The expectation is taken over

$$p_\theta(o_{\leq T}, \hat{s}_{\leq T}, a_{<T}) = p_D(o_{\leq T}, a_{<T}) p_\theta(\hat{s}_1 | o_1) \prod_{t=2}^{T} p_\theta(\hat{s}_t | \hat{s}_{t-1}, a_{t-1}, o_t)$$

where $p_D(o_{\leq T}, a_{<T})$ is the dataset distribution and $p_\theta(\hat{s}_1 | o_1)$ is the encoder $p_\theta(\hat{s}_t | \hat{s}_{t-1}, a_{t-1}, o_t)$ at the initial timestep. Note that the objective is equivalent to the ones in (Hafner et al., 2019; Lee et al., 2020).

## C  OBJECTIVE DERIVATION

### C.1  MAIN OBJECTIVE DERIVATION

We start by rewriting our objective in equation 3

$$J(\theta) = I_\theta(\hat{S}_t, \hat{E}_t; O_t) - I_\theta(\hat{S}_t; O_t | \hat{S}_{t-1}, A_{t-1}) - I_\theta(\hat{E}_t; O_t)$$

For the first term, the lower-bound can be derived as

$$I_\theta(\hat{S}_t, \hat{E}_t; O_t) = \mathbb{E}_{p_\theta(\hat{s}_t, \hat{e}_t, o_t)} \Big[ \log \frac{p_\theta(o_t | \hat{s}_t, \hat{e}_t)}{p(o_t)} \Big]$$

$$= \mathbb{E}_{p_\theta(\hat{s}_t, \hat{e}_t, o_t)} \Big[ \log \frac{p_\theta(o_t | \hat{s}_t, \hat{e}_t) q_\phi(o_t | \hat{s}_t, \hat{e}_t)}{p(o_t) q_\phi(o_t | \hat{s}_t, \hat{e}_t)} \Big]$$

$$= \mathbb{E}_{p_\theta(\hat{s}_t, \hat{e}_t, o_t)} \big[ \log q_\phi(o_t | \hat{s}_t, \hat{e}_t) \big] + H(O_t)$$

$$\quad + \mathbb{E}_{p_\theta(\hat{s}_t, \hat{e}_t)} \big[ D_{KL}(p_\theta(o_t | \hat{s}_t, \hat{e}_t) || q_\phi(o_t | \hat{s}_t, \hat{e}_t)) \big]$$

$$I_\theta(\hat{S}_t, \hat{E}_t; O_t) \geq \mathbb{E}_{p_\theta(\hat{s}_t, \hat{e}_t, o_t)} \big[ \log q_\phi(o_t | \hat{s}_t, \hat{e}_t) \big] + H(O_t) \tag{10}$$

For the second term, the lower-bound can be derived as

$$I_\theta(\hat{S}_t; O_t | \hat{S}_{t-1}, A_{t-1}) = \mathbb{E}_{p_\theta(\hat{s}_t, o_t, \hat{s}_{t-1}, \hat{a}_{t-1})} \Big[ \log \frac{p_\theta(\hat{s}_t | \hat{s}_{t-1}, a_{t-1}, o_t)}{p_\theta(\hat{s}_t | \hat{s}_{t-1}, a_{t-1})} \Big]$$

$$= \mathbb{E}_{p_\theta(\hat{s}_t, o_t, \hat{s}_{t-1}, \hat{a}_{t-1})} \Big[ \log \frac{p_\theta(\hat{s}_t | \hat{s}_{t-1}, a_{t-1}, o_t) q_\phi(\hat{s}_t | \hat{s}_{t-1}, a_{t-1})}{p_\theta(\hat{s}_t | \hat{s}_{t-1}, a_{t-1}) q_\phi(\hat{s}_t | \hat{s}_{t-1}, a_{t-1})} \Big]$$

$$= \mathbb{E}_{p_\theta(o_t, \hat{s}_{t-1}, \hat{a}_{t-1})} \big[ D_{KL}(p_\theta(\hat{s}_t | \hat{s}_{t-1}, a_{t-1}, o_t) || q_\phi(\hat{s}_t | \hat{s}_{t-1}, a_{t-1})) \big]$$

$$\quad - \mathbb{E}_{p_\theta(\hat{s}_{t-1}, \hat{a}_{t-1})} \big[ D_{KL}(q_\phi(\hat{s}_t | \hat{s}_{t-1}, a_{t-1}) || p_\theta(\hat{s}_t | \hat{s}_{t-1}, a_{t-1})) \big]$$

$$I_\theta(\hat{S}_t; O_t | \hat{S}_{t-1}, A_{t-1}) \leq \mathbb{E}_{p_\theta(o_t, \hat{s}_{t-1}, \hat{a}_{t-1})} \big[ D_{KL}(p_\theta(\hat{s}_t | \hat{s}_{t-1}, a_{t-1}, o_t) || q_\phi(\hat{s}_t | \hat{s}_{t-1}, a_{t-1})) \big]$$

$$-I_\theta(\hat{S}_t; O_t | \hat{S}_{t-1}, A_{t-1}) \geq -\mathbb{E}_{p_\theta(o_t, \hat{s}_{t-1}, \hat{a}_{t-1})} \big[ D_{KL}(p_\theta(\hat{s}_t | \hat{s}_{t-1}, a_{t-1}, o_t) || q_\phi(\hat{s}_t | \hat{s}_{t-1}, a_{t-1})) \big] \tag{11}$$

Using similar derivations as the second term, the third term is lower-bounded by

$$-I_\theta(\hat{E}_t; O_t | \hat{E}_{t-1}) \geq -\mathbb{E}_{p_\theta(o_t, \hat{e}_{t-1})} \big[ D_{KL}(p_\theta(\hat{e}_t | \hat{e}_{t-1}, o_t) || q_\phi(\hat{e}_t | \hat{e}_{t-1})) \big]$$

Thus, combining the three bounds, we can get $J_{\text{ELBO}}$ as in equation 5.

## C.2 Lower-Bound for Regularization Terms

The lower-bound for all the regularization terms as in equation 6 and equation 7 can be derived following similar techniques in equation 10.

## C.3 Practical Algorithm

Our encoders $p_\theta(\hat{s}_t|\hat{s}_{t-1}, a_{t-1}, o_t)$ and $p_\theta(\hat{e}_t|\hat{e}_{t-1}, o_t)$ require the inferred state and exogenous representations from previous timesteps. Following prior works on sequential latent variable models (Doerr et al., 2018; Lee et al., 2020), we expand the expectation of our objective in equation 8 to a sequence of length $T$ as follows

$$\mathbb{E}_{p_\theta(o_{\leq T}, \hat{s}_{\leq T}, \hat{e}_{\leq T}, a_{<T})}\Big[\sum_{t=1}^{T} \log q_\phi(o_t|\hat{s}_t, \hat{e}_t) + c_{\text{invdyn}} \sum_{t=1}^{T-1} \big(J_{\text{InvDyn-S}}(t; \theta, \phi) - J_{\text{InvDyn-E}}(t; \theta, \psi)\big)$$

$$- c_S \sum_{t=1}^{T} D_{KL}(p_\theta(\hat{s}_t|o_t, \hat{s}_{t-1}, a_{t-1})||q_\phi(\hat{s}_t|\hat{s}_{t-1}, a_{t-1})) - c_E \sum_{t=1}^{T} D_{KL}(p_\theta(\hat{e}_t|o_t, \hat{e}_{t-1})||q_\phi(\hat{e}_t))\Big].$$

The expectation is taken over

$$p_\theta(o_{\leq T}, \hat{s}_{\leq T}, \hat{e}_{\leq T}, a_{<T}) = p_D(o_{\leq T}, a_{<T}) p_\theta(\hat{s}_1|o_1) p_\theta(\hat{e}_1|o_1) \prod_{t=2}^{T} p_\theta(\hat{s}_t|\hat{s}_{t-1}, a_{t-1}, o_t) p_\theta(\hat{e}_t|\hat{e}_{t-1}, o_t)$$

where $p_D(o_{\leq T}, a_{<T})$ is the dataset distribution and $p_\theta(\hat{s}_1|o_1)$ and $p_\theta(\hat{e}_1|o_1)$ are the encoders $p_\theta(\hat{s}_t|\hat{s}_{t-1}, a_{t-1}, o_t)$ and $p_\theta(\hat{e}_t|\hat{e}_{t-1}, o_t)$ at the initial timestep, respectively.

# D  FULL EXPERIMENTAL RESULTS

In addition to the main results from Section 5, we add two additional baselines: (1) single-step inverse dynamics (**InvDyn**) and (2) **DINOv2**. Inverse dynamics has been shown to be effective for learning control-related features (Brandfonbrener et al., 2023) and DINOv2 (Oquab et al., 2024) is a powerful pre-trained image encoder.

| | | SLAC | TiA | Den-MDP | Iso-Dream | RePo | DrQ-v2 | InvDyn | ACRO | InfoGating | DINOv2 | CLEAR |
|---|---|---|---|---|---|---|---|---|---|---|---|---|
| Hopper | Clean (easy) | 88.2 ± 3.6 | 20.0 ± 2.8 | 25.9 ± 2.7 | 69.8 ± 7.5 | 4.5 ± 1.2 | 88.5 ± 2.6 | 52.9 ± 3.2 | 73.7 ± 3.9 | 78.1 ± 2.8 | 17.6 ± 2.9 | **104.9 ± 2.8** |
| | SV (medium) | 14.6 ± 1.2 | 1.9 ± 0.6 | 22.7 ± 3.0 | 28.0 ± 5.0 | 5.0 ± 1.0 | 64.1 ± 2.4 | 53.6 ± 2.1 | 64.0 ± 2.4 | **82.9 ± 2.1** | 2.7 ± 0.8 | 60.5 ± 5.1 |
| | MV (medium) | 4.6 ± 0.9 | 2.3 ± 0.3 | 10.0 ± 3.3 | 22.2 ± 7.2 | 3.9 ± 0.3 | 49.7 ± 2.0 | 44.4 ± 1.7 | 51.7 ± 1.5 | **62.0 ± 4.0** | 0.7 ± 0.2 | 39.8 ± 11.5 |
| | 2 × 2 (hard) | 5.4 ± 0.9 | 0.1 ± 0.1 | 8.3 ± 1.2 | 25.5 ± 4.6 | 3.6 ± 0.8 | 27.0 ± 3.9 | **44.8 ± 4.4** | 35.1 ± 3.1 | 44.7 ± 4.1 | 1.0 ± 0.3 | 50.5 ± 4.2 |
| Walker | Clean (easy) | 74.5 ± 11.6 | 79.6 ± 2.6 | 38.5 ± 3.2 | **83.5 ± 8.5** | 38.8 ± 3.2 | 75.6 ± 2.6 | **86.5 ± 1.9** | **89.7 ± 1.7** | **89.0 ± 1.1** | 36.4 ± 0.8 | 89.9 ± 2.0 |
| | SV (medium) | 79.9 ± 3.6 | 80.1 ± 2.7 | 50.9 ± 4.5 | **92.0 ± 1.3** | 35.8 ± 1.8 | 56.3 ± 1.7 | 82.9 ± 2.7 | 88.3 ± 1.0 | **90.7 ± 1.4** | 32.9 ± 1.7 | 87.6 ± 3.8 |
| | MV (medium) | 68.1 ± 1.8 | 62.8 ± 4.5 | 46.9 ± 2.3 | **84.3 ± 3.1** | 27.1 ± 5.9 | 62.3 ± 1.2 | 78.7 ± 1.3 | **88.8 ± 1.9** | 83.4 ± 3.5 | 27.4 ± 0.5 | 88.4 ± 1.8 |
| | 2 × 2 (hard) | 44.5 ± 3.8 | 26.5 ± 3.2 | 29.8 ± 3.0 | 80.1 ± 5.0 | 34.9 ± 3.3 | 45.7 ± 1.3 | 59.5 ± 1.9 | 76.4 ± 2.0 | 81.3 ± 2.5 | 18.8 ± 1.2 | **88.8 ± 2.4** |
| Cheetah | Clean (easy) | **95.0 ± 1.7** | 67.7 ± 6.2 | 43.6 ± 3.9 | 56.5 ± 12.7 | 38.1 ± 5.7 | 85.3 ± 3.2 | 31.5 ± 2.2 | 85.0 ± 3.1 | 72.5 ± 3.0 | 46.1 ± 2.9 | **96.5 ± 0.6** |
| | SV (medium) | 72.6 ± 4.2 | 58.9 ± 5.7 | 64.6 ± 3.9 | **94.5 ± 1.2** | 37.1 ± 4.4 | 73.6 ± 1.0 | 60.9 ± 4.1 | 79.9 ± 0.8 | 86.7 ± 1.8 | 28.7 ± 1.0 | **96.7 ± 1.5** |
| | MV (medium) | 54.7 ± 4.0 | 36.0 ± 3.7 | 45.5 ± 2.4 | **94.0 ± 3.1** | 43.1 ± 3.8 | 60.8 ± 2.5 | 38.8 ± 4.4 | 59.1 ± 3.5 | 68.1 ± 4.5 | 23.6 ± 1.4 | **95.8 ± 1.1** |
| | 2 × 2 (hard) | 46.2 ± 4.7 | 29.9 ± 1.3 | 39.0 ± 2.2 | 32.1 ± 3.0 | 37.7 ± 1.0 | 51.0 ± 2.7 | 27.7 ± 3.0 | 43.2 ± 1.6 | 47.4 ± 5.0 | 22.7 ± 10.0 | **79.1 ± 4.2** |

Table 4: Average normalized score and its standard error over 5 seeds on DeepMind Control Suite for Clean, Single Video (SV), Multiple Videos (MV) and $2 \times 2$ Grid distractions.

| | | SLAC | TiA | Den-MDP | Iso-Dream | RePo | InvDyn | ACRO | InfoGating | DINOv2 | CLEAR |
|---|---|---|---|---|---|---|---|---|---|---|---|
| Hopper | Clean (easy) | **0.92 ± 0.04** | 1.09 ± 0.05 | 1.30 ± 0.05 | **0.97 ± 0.09** | 1.92 ± 0.08 | 1.07 ± 0.01 | 1.08 ± 0.02 | 1.08 ± 0.03 | 1.22 | **1.04 ± 0.15** |
| | SV (medium) | 1.86 ± 0.07 | 2.79 ± 1.09 | **1.17 ± 0.06** | 1.35 ± 0.15 | 2.03 ± 0.05 | 1.33 ± 0.04 | 1.26 ± 0.04 | **1.18 ± 0.03** | 3.99 | **1.11 ± 0.10** |
| | MV (medium) | 2.94 ± 0.09 | 3.23 ± 1.39 | **1.93 ± 0.90** | 1.81 ± 0.71 | 2.08 ± 0.03 | 1.55 ± 0.04 | **1.43 ± 0.04** | **1.41 ± 0.06** | 6.02 | 1.58 ± 0.40 |
| | 2 × 2 (hard) | **1.26 ± 0.06** | 2.37 ± 0.11 | 1.69 ± 0.10 | 1.36 ± 0.12 | 2.04 ± 0.10 | 1.59 ± 0.06 | 1.55 ± 0.06 | 1.54 ± 0.06 | 2.78 | **1.15 ± 0.09** |
| Walker | Clean (easy) | **2.59 ± 0.04** | 2.92 ± 0.05 | 2.79 ± 0.04 | **2.72 ± 0.19** | 4.17 ± 0.07 | 3.37 ± 0.05 | 3.49 ± 0.03 | 3.38 ± 0.08 | 3.91 | **2.62 ± 0.06** |
| | SV (medium) | 3.55 ± 0.18 | 4.01 ± 0.24 | 2.89 ± 0.07 | **2.52 ± 0.03** | 4.21 ± 0.06 | 3.62 ± 0.07 | 3.69 ± 0.06 | 3.60 ± 0.05 | 5.86 | 2.99 ± 0.25 |
| | MV (medium) | 3.91 ± 0.22 | 4.19 ± 0.24 | **2.87 ± 0.09** | **2.76 ± 0.19** | 4.20 ± 0.03 | 3.77 ± 0.04 | 3.86 ± 0.08 | 3.70 ± 0.06 | 6.68 | **3.04 ± 0.32** |
| | 2 × 2 (hard) | 4.45 ± 0.09 | 5.92 ± 0.12 | 3.93 ± 0.13 | 4.15 ± 0.33 | 4.30 ± 0.12 | 4.33 ± 0.06 | 4.33 ± 0.07 | 4.23 ± 0.14 | 7.19 | **3.27 ± 0.07** |
| Cheetah | Clean (easy) | **0.83 ± 0.02** | 1.21 ± 0.06 | 1.62 ± 0.04 | **0.85 ± 0.01** | 2.52 ± 0.05 | 1.42 ± 0.04 | 1.81 ± 0.04 | 1.83 ± 0.05 | 3.47 | **0.88 ± 0.03** |
| | SV (medium) | 3.08 ± 0.11 | 4.08 ± 1.11 | 2.97 ± 0.20 | **1.51 ± 0.26** | 2.70 ± 0.11 | 2.16 ± 0.03 | 2.44 ± 0.05 | 2.37 ± 0.04 | 8.14 | **1.22 ± 0.56** |
| | MV (medium) | 4.07 ± 0.06 | 5.34 ± 0.47 | 3.00 ± 0.29 | **1.29 ± 0.16** | 2.58 ± 0.07 | 2.63 ± 0.05 | 2.81 ± 0.07 | 2.78 ± 0.06 | 11.45 | **1.19 ± 0.22** |
| | 2 × 2 (hard) | 1.29 ± 0.02 | 1.76 ± 0.01 | 1.60 ± 0.06 | 1.27 ± 0.02 | 2.75 ± 0.06 | 2.23 ± 0.04 | 2.37 ± 0.03 | 2.37 ± 0.12 | 5.63 | **1.14 ± 0.04** |

Table 5: Average MSE and its standard deviation over 5 seeds on the ground-truth state regression task using linear model.

From the results in Table 4, we see that even powerful image encoders such as DINOv2 performs poorly in offline RL. This is due to the fact that these methods are not regularized to remove any information about uncontrollable distractions, which is a problem specific to RL. We also note that for all environments, the performance consistently decreases as the background distractions become more complex.

# E EVALUATION ON UNSEEN BACKGROUND

In addition to our main results, we also evaluate CLEAR's ability to generalize to unseen background distractions. When the background distractions are different from that of the training dataset, this introduces an additional problem of distribution shift, which is a common challenge in many machine learning problems. More specifically, the data distribution $p_D$ used to optimize the objective (see equation 8) shifts in the case of novel unseen backgrounds.

|  | Videos | Iso-Dream | InfoGating | CLEAR |
|---|---|---|---|---|
| Walker | 1 | $21.7 \pm 4.1$ | $\mathbf{79.7 \pm 3.3}$ | $\mathbf{80.2 \pm 0.1}$ |
|  | 4 | $\mathbf{85.2 \pm 3.0}$ | $\mathbf{88.1 \pm 1.6}$ | $82.2 \pm 6.3$ |
|  | 10 | $\mathbf{88.1 \pm 2.5}$ | $\mathbf{88.3 \pm 1.6}$ | $86.8 \pm 2.2$ |
|  | 25 | $\mathbf{90.1 \pm 1.0}$ | $\mathbf{90.8 \pm 1.3}$ | $88.5 \pm 2.2$ |
| Cheetah | 1 | $4.5 \pm 1.7$ | $24.2 \pm 5.1$ | $\mathbf{71.1 \pm 3.2}$ |
|  | 4 | $45.5 \pm 2.2$ | $65.1 \pm 3.2$ | $\mathbf{90.8 \pm 1.2}$ |
|  | 10 | $53.7 \pm 5.9$ | $57.8 \pm 5.3$ | $\mathbf{91.5 \pm 1.1}$ |
|  | 25 | $64.0 \pm 9.6$ | $53.2 \pm 4.5$ | $\mathbf{92.7 \pm 1.9}$ |

Table 6: Average normalized score and its standard error over 5 seeds on unseen background videos.

In Table 6, we show the results for training on 1, 4, 10 and 25 different video distractions and testing on the unseen video backgrounds. Intuitively, training on a larger number of background distractions covers a "wider" distribution for $p_D$. We add random convolutions, which is a well-known heuristic to handle distribution shift. Specifically, we apply random convolution to the image fed into the state encoder $p_\theta(\hat{s}_t | \hat{s}_{t-1}, a_{t-1}, o_t)$ during pretraining and offline RL training. We can see that CLEAR outperforms the strongest baselines when evaluated on novel unseen backgrounds specifically on Cheetah environment. The results also show the tendency of increasing performance as the number of background videos seen in the dataset increase. It shows that as $p_D$ covers a "wider" distribution, generalization to unseen background improves as well.

## F    EXPERIMENT ON ADDITIONAL ENVIRONMENT

|         |              | InfoGating      | CLEAR         |
|---------|--------------|-----------------|---------------|
| Finger  | Clean (`easy`)   | 101.7 ± 0.2  | 99.7 ± 1.3    |
|         | SV (`medium`)    | 101.2 ± 0.1  | 101.0 ± 0.2   |
|         | MV (`medium`)    | 100.4 ± 0.3  | 98.1 ± 1.3    |
|         | 2 × 2 (`hard`)   | 100.3 ± 0.2  | 99.6 ± 0.8    |
| Cartpole | Clean (`easy`)  | 0.9 ± 0.0    | 87.3 ± 4.7    |
|         | SV (`medium`)    | 0.9 ± 0.0    | 58.7 ± 8.1    |
|         | MV (`medium`)    | 0.9 ± 0.0    | 39.3 ± 6.4    |
|         | 2 × 2 (`hard`)   | 0.9 ± 0.0    | 42.1 ± 1.1    |

Table 7: Average normalized score and its standard error over 3 seeds.

Table 7 shows the result on an additional environment on DeepMind Control Suite called Finger-Spin and Cartpole-Swingup, which was used in previous works (Zhu et al., 2023; Bharadhwaj et al., 2022). We would like to reiterate that normalized score of 100 means that the representation learned is as good as if we have access to the ground-truth state. Thus, we note that on the simpler Finger environment, CLEAR and InfoGating (the strongest baselines) already performed optimally as if they have access to the ground-truth state.

On the other hand, InfoGating fails to learn on seemingly simple Cartpole environment because of how it handles partial observability. InfoGating stacks consecutive frames and use it as its input. In the Cartpole environment, the agent may disappear from the screen, making it impossible to determine its ground-truth state by stacking consecutive frames. In the other hand, CLEAR handles partial observability by learning latent state dynamics which can track its state over time and handle such case.

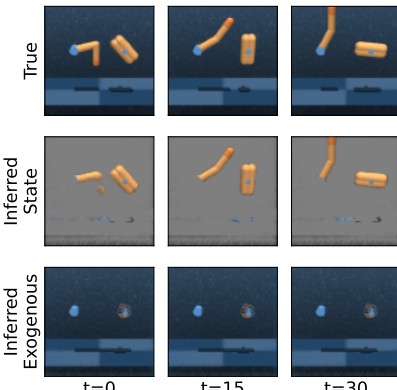

Figure 7: Qualitative results of CLEAR for Finger-Spin dataset.

Interestingly, as opposed to Figure 4 where the floor is part of the inferred state since it correlates with the controllable state, it is not the case with Finger environment as shown in Figure 7. We characterize distraction as a factor that 1) does not affect the reward function, 2) is unaffected by action, 3) is independent of the agent state, and 4) is present in the observation. As such, the floor in Finger environment is independent of the agent state. Notably, while the right-hand side body (the part being spinned) is not directly actuated, it is still part of the inferred state since it is dependent on the agent state.

# G    DATASET DETAILS

We train a medium policy and an expert policy using state-based SAC (Haarnoja et al., 2018) as opposed to image-based RL methods to generate the dataset. Using image-based RL methods as the data collecting policy will bias the dataset towards being easy under vision-based methods (Lu et al., 2023). Using the trained policies, we can rollout the policies and render the image during rollout to generate the dataset. We use three environments from DeepMind Control Suite (Tassa et al., 2018): Hopper-Hop, Walker-Walk, and Cheetah-Run. The medium policy is trained for 250k timesteps in Hopper-Hop, 200k timesteps in Walker-Walk, and 400k timesteps in Cheetah-Run. For the expert policy, it is trained for 1M timesteps in Walker-Walk as well as Cheetah-Run and 2M timesteps in Hopper-Hop. Using the trained policies, we collect a dataset of 200 episodes where each episode is 500 timesteps (frame-skip of 2). The dataset statistics are provided in Table 8.

| Dataset | | Timesteps | Mean | Std. Dev. | Min. | Max. |
|---------|--------|-----------|--------|-----------|--------|--------|
| hopper-hop | medium | 100k | 185.17 | 21.28 | 0.0 | 206.85 |
| | expert | 100k | 309.07 | 31.58 | 0.0 | 326.15 |
| walker-walk | medium | 100k | 587.26 | 35.18 | 471.23 | 644.91 |
| | expert | 100k | 957.04 | 17.63 | 812.76 | 986.41 |
| cheetah-run | medium | 100k | 477.05 | 84.93 | 102.91 | 573.63 |
| | expert | 100k | 748.40 | 11.27 | 718.87 | 775.97 |

Table 8: The statistics of collected dataset that are used in our main experiments. The mean, standard deviation, minimum, and maximum are the statistics of the returns in the dataset.

For each environment, we combine the medium and expert datasets to make a medium-expert dataset. From this dataset, we render different types of background distractions so that we can isolate the effect of different distractions on the policy performance. Thus, an optimal representation learning algorithm should perform equally on the same environment across different level of distractions. We use an image size of $84 \times 84$. Our implementation of the background distraction relies on the Distracting Control Suite (Stone et al., 2021). We generate four different distractions which we will explain below, starting from the easiest to the hardest distraction level. We provide samples of the dataset in Figure 8.

1. **Clean**. We do not modify the background. Since the color of the agent contrasts with the background, the algorithm may rely on color to extract control-related features.

2. **Single Video (SV)**. We use video as background distraction throughout an episode. This is a harder setting since the distraction is time-correlated. We only use a single video. However, when we reset the environment, the starting frame of the video might be different across episodes. Since the number of frames of the video is less than the number of frames per episode of the environment, we reverse the video when it reaches the end or the beginning.

3. **Multiple Videos (MV)**. This is similar to the **Single Video** setting. However, when we reset the environment, we not only change the starting frame but also the video. We use four videos in this setting.

4. **2 × 2 Grid**. First we render the agent similar to the **Clean** setting. Then, we downsize the image to $41 \times 41$ to place it on the top-left position of a $2 \times 2$ grid. The rest of the grid is filled with the same agent that we are trying to control but are not controllable. These uncontrollable agents are generated by random policy.

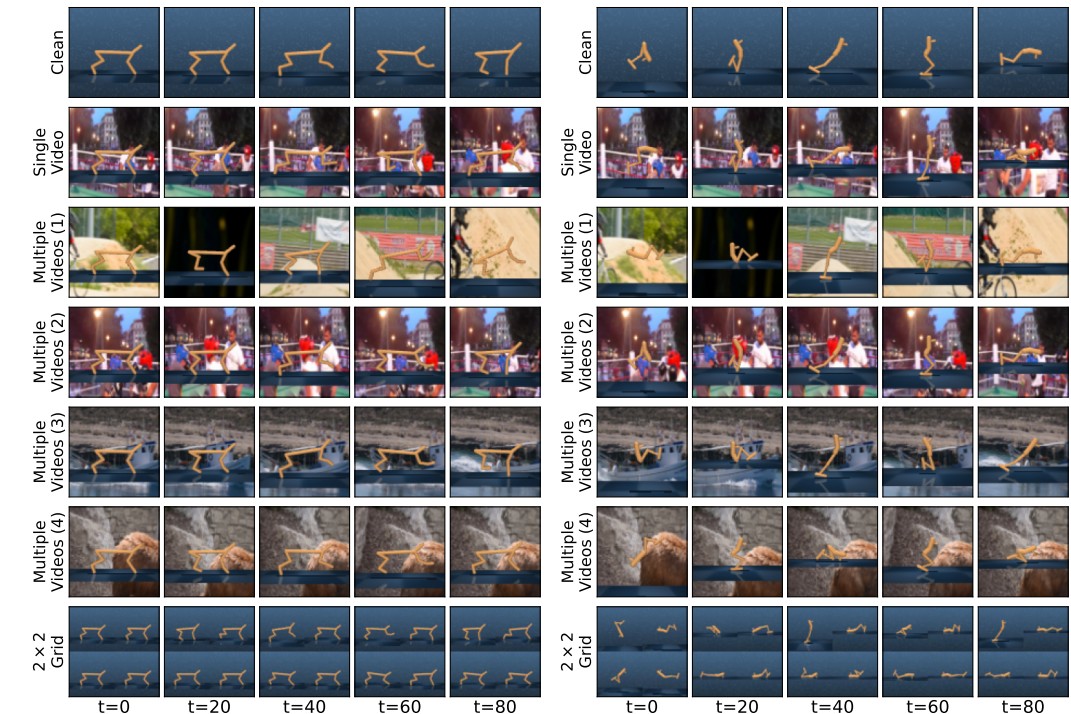

Figure 8: Sample of dataset with different distractions that we use in our main experiment for Cheetah-Run and Hopper-Hop environments.

## H  HYPERPARAMETERS AND BASELINES

### H.1  OFFLINE RL: TD3 + BC

We run TD3+BC (Fujimoto & Gu, 2021) as the offline RL algorithm on top of the learned representation. We freeze all the pretrained encoders during the offline RL training since the dataset is fixed. The objective of TD3+BC is to optimize TD3 (Fujimoto et al., 2018) as well as BC objective as

$$\pi = \arg\max_{\pi} \mathbb{E}_{(s,a)\sim D}\Big[\lambda Q(s, \pi(s)) - (\pi(s) - a)^2\Big]$$

with

$$\lambda = \frac{\alpha}{\frac{1}{N}\sum_{s_i,a_i}|Q(s_i, a_i)|}$$

where $\alpha$ is a non-negative number as a hyperparameter and $s$ is either the ground-truh state in case of state-based RL or the output of the pretrained encoder in case of image-based RL. Following prior works (Fujimoto & Gu, 2021; Lu et al., 2023), we pick 2.5 as the value of $\alpha$. Table 9 shows the results of running TD3+BC on the ground-truth state. We use the mean of the return to normalize the reported score of all methods.

| Environment | Return |
|---|---|
| Hopper | 188.0 ± 19.2 |
| Walker | 953.3 ± 3.1 |
| Cheetah | 776.0 ± 2.9 |

Table 9: Results of running TD3+BC on the ground-truth state over 5 seeds using $\alpha = 2.5$. The reported score is the mean and standard error of the return.

## H.2 FRAME-STACKING APPROACHES: DRQ-V2, INVDYN, ACRO, INFOGATING, DINOV2

For the frame-stacking approaches, we follow their respective original implementations (Yarats et al., 2022; Lu et al., 2023; Islam et al., 2023). We 1) stack 3 consecutive image observations, 2) use $N$-step return for the bootstrapped target with $N = 3$, and 3) apply cropping-based augmentation (Yarats et al., 2021). DrQ-v2 has no pretraining step thus the encoder is not frozen during offline RL training. The encoders of ACRO and InvDyn are frozen during offline RL training and are trained to optimize the following objective during pretraining

$$\max_{\theta,\phi} \mathbb{E}_{k \sim U(1,K),(o_t,a_t,o_{t+k}) \sim D} \left[ \log p_\theta(a_t | \phi(o_t), \phi(o_{t+k})) \right]$$

where $U(1, K)$ is a uniform distribution over $\{1, 2, ..., K\}$, $a$ is the action, $o$ is the stacked image observation, $\phi$ is the encoder, $\theta$ is the action predictor which is not used during offline RL training, and $K \in \mathbb{N}$ is a hyerparameter. For ACRO, we searched $K$ between $\{8, 15\}$ and found that $K = 15$ is the best one as reported by the paper (Islam et al., 2023), while $K = 1$ is set for InvDyn. We apply image augmentation during offline RL and pretraining since we observe that, without image augmentation, ACRO performs significantly worse. For InfoGating, an extension of ACRO, we used the official repository and tuned the hyperparameter $\lambda \in 1, 0.1, 0.01$, which balances the L1-loss for mask learning and the multi-step inverse dynamics loss. We found $\lambda = 0.01$ to be optimal, and all reported results are based on this value. For DINOv2 (Oquab et al., 2024), we stack three consecutive representations (i.e. the class token) as an input to the offline RL agent. We use dinov2_vits14 model from the official repository.

## H.3 LATENT DYNAMICS APPROACHES: SLAC, ISO-DREAM, TIA, DENOISEDMDP, REPO

Table 10 shows the hyperparameters that we use for the reported score in the main experiments.

**SLAC** (Lee et al., 2020) models a single latent variable and optimizes its ELBO with additional reward prediction. We follow the implementation of SLAC by factorizing the variable as explained in the Appendix B of the paper. The latent variables have 32 and 256 dimensions for $z_1$ and $z_2$, respectively. The decoder $q_\phi(o_t | \hat{s}_t)$ is parameterized as an independent Gaussian for each pixel whose variance is fixed to a constant. We search the variance between $\{0.4, 0.1, 0.04\}$.

**TiA** (Fu et al., 2021) models two latent variables and regularize it via reward prediction. However, their method assumes both latent variables are controllable (i.e. affected by action). Additionally, reward prediction makes the learned representation to be task-dependent and is problematic since reward may be sparse or depends only on the subset of the agent state. TiA has two hyperparameters, namely $\lambda_{\text{Radv}}$ which controls the adversarial reward regularizer and $\lambda_{O_s}$ which controls distractor-model-only reconstruction. We refer readers to (Fu et al., 2021) for the details of the objective. We follow the original implementation which parameterizes both encoders as RSSM (Hafner et al., 2019). We pick 30 and 200 dimensions for the stochastic and deterministic variables, respectively. We search $\lambda_{\text{Radv}}$ between $\{20k, 30k\}$ and $\lambda_{O_s}$ between $\{0.25, 1.5, 2.0\}$.

**DenoisedMDP** (Wang et al., 2022) models multiple latent variables based on its controllability and task relevance. We use the official implementation which uses the Figure 2b variant of the paper. It has two hyperparameters, $\alpha$ which weights the KL divergence of the controllable representation and $\beta$ which weights the KL divergence of the rest. We search over $\{1., 2.\}$ for $\alpha$ and $\{1., 0.5, 0.25, 0.125\}$ for $\beta$.

**Iso-Dream** (Pan et al., 2022) models three latent variables with a single regularization (inverse dynamics prediction) on the controllable representation. We set all KL weights to 1, following the paper's reported hyperparameters. However, we tuned the image decoder variance and found the optimal hyperparameters to match those used in CLEAR (ours).

**RePo** (Zhu et al., 2023) models a single latent variable and avoids observation reconstruction altogether. Instead, it reconstructs reward to extract task-relevant information. The issue is similar with TiA since reward may be sparse or depends only on the subset of the agent state. Similar to TiA, we use RSSM as the encoder with 30 and 200 dimensions for the stochastic and deterministic variables, respectively. In the original implementation, the weight between reward prediction and KL divergence is learned. However, we found that it does not perform well. Instead, we search over $\{1, 10^{-1}, 10^{-2}, 10^{-3}, 10^{-3}, 10^{-4}, 10^{-5}, 10^{-6}\}$ for the KL weight and $\{1., 0.1, 0.04, 0.01\}$ for the variance of reward predictor. However, we still found that none of them work well. We argue that this is

due to finite dataset and the reward is uninformative to learn a meaningful representation which has been demonstrated in (Hafner et al., 2020). Nonetheless, we report the score of the most performing ones with $10^{-5}$ as the KL divergence weight and $0.04$ as the variance of the reward predictor.

**CLEAR** optimizes the regularized objective in Eq equation 8. For encoder $p_\theta(\hat{s}_t|\hat{s}_{t-1}, a_{t-1}, o_t)$, we use RSSM with 30 and 200 dimensions for the stochastic and deterministic variables, respectively. Likewise, for encoder $p_\theta(\hat{e}_t|o_t)$, we use 30 dimensions. We also follow SLAC where we model the decoder with a fixed variance $\sigma^2$. For inverse dynamics prediction, we set the variance of the output to be 0.002 for all experiments and search over $c_S$ and $c_E$ which are the weights for KL divergence of state and exogenous variables, respectively. For the Clean dataset, we found that just setting $c_S = c_E = 1.0$ works fine and search $\sigma^2$ over $\{0.1, 0.02, 0.04\}$. For the Videos dataset, we search the $c_E$ to be among $\{1.0, 0.1\}$ since now the distractions present. Lastly, for $2 \times 2$ Grid, we further extend the search since not only now distractions present, but also the size of the controllable agent is now smaller in the image observation.

| | | SLAC $\sigma^2$ | TiA $\lambda_{\text{Radv}}, \lambda_{O_s}$ | DenoisedMDP $\alpha, \beta$ | CLEAR $\sigma^2, c_S, c_E$ |
|---|---|---|---|---|---|
| Hopper | Clean (`easy`) | 0.04 | 20k, 2.00 | 2., 1.00 | 0.02, 1.0, 1.0 |
| | SV (`medium`) | 0.04 | 20k, 2.00 | 1., 0.50 | 0.04, 1.0, 1.0 |
| | MV (`medium`) | 0.04 | 20k, 2.00 | 1., 0.50 | 0.04, 1.0, 0.1 |
| | $2 \times 2$ (`hard`) | 0.04 | 20k, 2.00 | 2., 0.50 | 0.01, 1.0, 0.5 |
| Walker | Clean (`easy`) | 0.04 | 30k, 0.25 | 1., 1.00 | 0.10, 1.0, 1.0 |
| | SV (`medium`) | 0.10 | 20k, 0.25 | 1., 0.50 | 0.10, 1.0, 0.1 |
| | MV (`medium`) | 0.10 | 20k, 0.25 | 1., 0.50 | 0.10, 1.0, 0.1 |
| | $2 \times 2$ (`hard`) | 0.04 | 30k, 1.50 | 1., 0.25 | 0.02, 2.0, 1.0 |
| Cheetah | Clean (`easy`) | 0.10 | 20k, 2.00 | 1., 0.50 | 0.10, 1.0, 1.0 |
| | SV (`medium`) | 0.10 | 20k, 2.00 | 1., 0.25 | 0.10, 1.0, 0.1 |
| | MV (`medium`) | 0.10 | 20k, 2.00 | 1., 0.25 | 0.10, 1.0, 0.1 |
| | $2 \times 2$ (`hard`) | 0.04 | 20k, 0.25 | 2., 0.50 | 0.01, 1.0, 0.5 |

Table 10: Hyperparamters for SLAC, TiA, DenoisedMDP, and CLEAR.

