# OpenReview forum: "CLEAR: An Information-Theoretic  Framework for Distraction-Free Representation Learning in Visual Offline RL"
_ICLR.cc/2025/Conference — Submitted to ICLR 2025_

### Official Review · Reviewer_UDWw · 2024-10-24

**Soundness:** 3
**Presentation:** 3
**Contribution:** 3
**Rating:** 5
**Confidence:** 3

**Summary:**

This paper proposes CLEAR, an agent-centric state-representation method for vision-based offline reinforcement learning. The authors present an information theoretical approach to achieve this goal, disentangling exogenous variables while enforcing the Markovian properties. For computation, the author proposed a lower bound of compounded mutual information in Eq. (3) to  Eq. (4) and designed a corresponding encoder-decoder architecture that can be trained with image representation techniques. The authors demonstrated that learned state representations by the proposed CLEAR method show good performance when there is uncontrollable visual distraction in offline RL settings.

**Strengths:**

1. The paper is well-written. It was straightforward to understand the core idea; despite dealing with complex concepts. Figures and equations are appropriately placed.
2. The proposed method is sound and properly designed to solve the offline RL problems.
3. In the visual domain, circumstances that ExoPOMDP describes are very common. This work can directly contribute to solving RL in real-world problems.
4. The authors have clearly put significant effort into reducing computation costs and simplifying architecture design. The work focus on enhancing efficiency and practicality of information theoretical approaches. I think most of the results could be implemented and reproduced even though source code is not provided.

**Weaknesses:**

1. Although the authors' claim primarily in the visual offline RL, the conducted experiments are strictly limited to MuJoCo variants. The authors are strongly encouraged to strengthen the experiment section by including (pure) vision-based RL problems with distractions.
2. Extensibility. I think the adaptation ability of CLEAR representation for different visual distractions from the training environments is only partially validated by multiple videos. The authors are encouraged to analyze general extensibility of CLEAR in these aspects for real-world scenarios.
3. It would be better if intuitive motivation or a toy experiment empirically verifies the ExoPOMDP setting.

**Questions:**

1. In the graphical model in Fig 2, what happens when e_t is affected by s_{t-1}? Does this modification change Eqs. (2) and (3)?

---

> ### Author Response · Authors · 2024-11-22
>
> # Response to Reviewer $\text{\textcolor{green}{UDWw}}$
>
> Thank you for taking the time to review our paper. Please let us know if there are further questions or necessary clarifications and we would be happy to address them during the discussion period.
>
> > **W1. Although the authors' claim primarily in the visual offline RL, the conducted experiments are strictly limited to MuJoCo variants. The authors are strongly encouraged to strengthen the experiment section by including (pure) vision-based RL problems with distractions.**
>
> Regarding the benchmarks that we have considered, we closely follow previous work in visual offline RL [1, 2] where DeepMind Control Suite is considered to be the standard benchmark. Our setting is purely vision-based RL problems, using only image observations without access to ground-truth states, with added distractions. We recommend the reviewer refer to Section 2.1 and the Datasets subsection in Section 5. Additionally, we have included results from other environments.
>
> | Task | | &nbsp;InfoGating&nbsp; | &nbsp;&nbsp;&nbsp;CLEAR |
> |-|:-:|:-:|:-:|
> | Finger | Clean (easy) | 101.7 ± 0.2 | 99.7 ± 1.3 |
> | | SV (medium) | 101.2 ± 0.1 | 101.0 ± 0.2 |
> | | MV (medium)| 100.4 ± 0.3 | 98.1 ± 1.3 |
> | | 2 × 2 (hard) | 100.3 ± 0.2 | 99.6 ± 0.8 |
> | Cartpole | Clean (easy) | 0.9 ± 0.0 | 87.3 ± 4.7 |
> | | SV (medium) | 0.9 ± 0.0 | 58.7 ± 8.1 |
> | | MV (medium) | 0.9 ± 0.0 | 39.3 ± 6.4 |
> | | 2 × 2 (hard) | 0.9 ± 0.0 | 42.1 ± 1.1 |
>
> The table shows the result on an additional environment on DeepMind Control Suite called Finger-Spin and Cartpole-Swingup, which were used in previous works [3, 4]. We refer the reviewer to take a look at Appendix F for detailed analysis.
>
> > **W2. Extensibility. I think the adaptation ability of CLEAR representation for different visual distractions from the training environments is only partially validated by multiple videos. The authors are encouraged to analyze general extensibility of CLEAR in these aspects for real-world scenarios.**
>
> We strongly believe that the multiple video distraction scenario well covers the generalization of CLEAR representation. In Appendix E, we have shown that unseen distractions can be viewed as a problem of distribution shift, which can be mitigated by, for instance, random convolutions. Thus, we view any alternative approach which can mitigate this problem of distribution shift as compatible with our framework. We note that one of the underlying assumptions with our method as well as baselines is that the presentation of the agent remains the same while the distractions can change. If the reviewer is asking, for example, to test the CLEAR encoder on 2x2 while trained on Single Video, this breaks the aforementioned assumption since the presentation of the agent is different in those two scenarios (agent size and location in images). We ask the reviewer to provide a detailed suggestion on alternative experimental setups.
>
> > **W3. It would be better if intuitive motivation or a toy experiment empirically verifies the ExoPOMDP setting.**
>
> We have provided an intuitive real world example in which visual observations often contain complex distractions, such as background screens playing video advertisements or birds flying in the sky, which are irrelevant to the control task at hand. Please see Figure 1 in the paper for further illustrations.
>
> > **Q1. In the graphical model in Fig 2, what happens when $e_t$ is affected by $s_{t-1}$? Does this modification change Eqs. (2) and (3)?**
>
> We characterize distraction as a factor that 1) does not affect the reward function, 2) is unaffected by action, 3) is independent of the agent state, and 4) is present in the observation. When $e_t$ is affected by $s_{t-1}$, we can no longer call the distraction $E$ independent of agent state $S$, violating the definition of distraction. Thus, we can no longer call it distraction $E$ but rather part of the agent state $S$ and the ExoPOMDP reduces to a POMDP.
>
> To give an example, consider the floor part of the visual observation (Figure 4). Although the floor is uncontrollable, the inferred state captures the floor as a controllable part. This is because how the floor moves depends on how the agent moves. We also make an additional example where the floor does not correlate with the agent state in the Finger environment (Appendix F in the updated manuscript). In such a case, the floor is inferred as part of the exogenous variables.
>
> > **S1. I think most of the results could be implemented and reproduced even though source code is not provided.**
>
> We would like to correct the reviewer that we have provided the source code in https://anonymous.4open.science/r/anonymous-clear-EFFC as we put in footnote 2 (Line 375)

---

> ### Author Response · Authors · 2024-11-22
>
> **References**
>
> [1] Challenges and Opportunities in Offline Reinforcement Learning from Visual Observations (Lu et, al. TMLR 2023)
>
> [2] Principled Offline RL in the Presence of Rich Exogenous Information (Islam et, al. ICML 2023)
>
> [3] "Information prioritization through empowerment in visual model-based RL." Bharadhwaj et al. (ICLR 2022)
>
> [4] “RePo: Resilient Model-Based Reinforcement Learning by Regularizing Posterior Predictability” Zhu et al. (NeurIPS 2023)

---

> > ### Comment · Reviewer_UDWw · 2024-11-25
> > **Response to Rebuttal**
> >
> > Thank you for your response and detailed explanation. I will finalize my score after discussion with other reviewers.

---

> > > ### Author Response · Authors · 2024-11-26
> > >
> > > Thank you again for taking the time to provide feedback on our work. Please let us know if there are further questions or necessary clarifications and we would be happy to address them during the discussion period.

---

> > > > ### Author Response · Authors · 2024-12-03
> > > >
> > > > With the deadline approaching in less than 8 hours, we would like to remind you that we are happy to promptly address any remaining concerns. Additionally, we kindly ask you to take into account our discussion with Reviewer $\text{\textcolor{blue}{UCLi}}$ as we believe this will be helpful for finalizing your evaluation of our paper.
> > > >
> > > > Thank you for your time and consideration.

---

### Official Review · Reviewer_UCLi · 2024-10-31

**Soundness:** 3
**Presentation:** 3
**Contribution:** 3
**Rating:** 6
**Confidence:** 3

**Summary:**

The paper addresses the problem of visual offline reinforcement learning in settings with distractions in visual observations. The authors formalize this setting as an ExoPOMDP and propose CLEAR, a method that learns the dynamics model of a agent-centric state representation consistent with the ExoPOMDP framework. Specifically, they take an information-theoretic approach, introducing mutual information-based loss functions to disentangle distraction factors from the agent-centric representation. Their method is evaluated on three tasks in the DeepMind Control Suite with diverse types of distractions.

**Strengths:**

1/ The high-level motivation of the method and the information-theoretic foundation look solid. Specifically, the regularization term that encourages s_hat to encode an agent-centric state controllable by actions is particularly interesting, and its effectiveness is well supported by the experiments.

2/ They evaluate the methods on various types of distractions, with 2x2 grid distractions being especially notable. This setup serves as a great testbed for assessing whether a method can distinguish controllable parts from non-controllable ones. The experiments show that their method consistently outperforms other methods in this setting, indicating that the baselines lack the ability to discern controllability, while their method succeeds.

3/ Their experiments include a comprehensive set of prior work as baselines, covering not only offline RL but also model-based RL. This strengthens the paper by demonstrating that their method outperforms approaches across various frameworks.

**Weaknesses:**

1/ Three tasks from a single domain seem too few, particularly since some baselines are comparable to the proposed method in some tasks (e.g. InfoGating on Hopper Hop). Expanding the evaluation to include more tasks (e.g., 5-6 tasks) would significantly strengthen the paper. A larger set of tasks would provide a more robust assessment of the method's performance and demonstrate its versatility across different scenarios.

2/ Although a large set of prior works is presented as baselines, some important and closely related studies are missing. Notable examples include [1] and [2]. [1] also employs information theory to address controllability. A detailed comparison between [1] and CLEAR is necessary, both methodologically and experimentally. [2] leverages causality to learn disentangled representations across four different categories, focusing on controllability and reward relevance.

[1] Homanga Bharadhwaj, et al. "Information prioritization through empowerment in visual model-based RL." ICLR 2022

[2] Yuren Liu, et al. "Learning world models with identifiable factorization." Neurips 2023

**Questions:**

1/ Which representation is used for offline RL (i.e., policy learning)? I assume s_hat is used and not e_hat, but I would like to clarify this.

2/ The method requires extensive hyperparameter search over the variance and balancing factors of the KL divergence for each task, domain, and distraction type (Table 9). It would be much more convenient to eliminate this necessity. Do you have any ideas on how to achieve this?

3/ Since the method aims to improve representation learning, it could potentially be applied within an MBRL framework. How do you think it would perform when integrated with an MBRL framework?

4/ How can this method handle other forms of nuisance information, such as color perturbations of task-relevant parts, where the assumption in L278 does not hold?

5/ Including more detailed descriptions of each column in the caption of Table 3 would make it much easier to interpret the results.

---

> ### Author Response · Authors · 2024-11-22
>
> # Response to Reviewer $\text{\textcolor{blue}{UCLi}}$
>
> Thank you for taking the time to review our submission and providing constructive feedback. We address each concern one-by-one below. Please let us know if there are further questions or necessary clarifications and we would be happy to address them during the discussion period.
>
> > **W1. Three tasks from a single domain seem too few, particularly since some baselines are comparable to the proposed method in some tasks (e.g. InfoGating on Hopper Hop). Expanding the evaluation to include more tasks (e.g., 5-6 tasks) would significantly strengthen the paper. A larger set of tasks would provide a more robust assessment of the method's performance and demonstrate its versatility across different scenarios.**
>
> We have provided results on additional environment(s) on Appendix F in the updated manuscript.
>
> | Task | | &nbsp;InfoGating&nbsp; | &nbsp;&nbsp;&nbsp;CLEAR |
> |-|:-:|:-:|:-:|
> | Finger | Clean (easy) | 101.7 ± 0.2 | 99.7 ± 1.3 |
> | | SV (medium) | 101.2 ± 0.1 | 101.0 ± 0.2 |
> | | MV (medium)| 100.4 ± 0.3 | 98.1 ± 1.3 |
> | | 2 × 2 (hard) | 100.3 ± 0.2 | 99.6 ± 0.8 |
> | Cartpole | Clean (easy) | 0.9 ± 0.0 | 87.3 ± 4.7 |
> | | SV (medium) | 0.9 ± 0.0 | 58.7 ± 8.1 |
> | | MV (medium) | 0.9 ± 0.0 | 39.3 ± 6.4 |
> | | 2 × 2 (hard) | 0.9 ± 0.0 | 42.1 ± 1.1 |
>
> The table shows the result on an additional environment on DeepMind Control Suite called Finger-Spin and Cartpole-Swingup, which were used in previous works [1, 3]. We refer the reviewer to take a look at Appendix F for detailed analysis.
>
> > **W2. Although a large set of prior works is presented as baselines, some important and closely related studies are missing. Notable examples include [1] and [2]. [1] also employs information theory to address controllability. A detailed comparison between [1] and CLEAR is necessary, both methodologically and experimentally. [2] leverages causality to learn disentangled representations across four different categories, focusing on controllability and reward relevance.**
>
> We have incorporated the suggested baseline, InfoPower [1], in the updated manuscript (Table 1 and Table 2). Since its source code is unavailable, we reimplemented the method to the best of our effort given the limited details on the paper. We verified our implementation by checking that the ground-truth state regression yielded values within a reasonable range. Despite having relatively low MSE on ground-truth state regression (Table 2), the offline RL result for all environments and distractions are significantly lower than that of CLEAR as well as other baselines (Table 1). We argue that this is due to the contrastive loss which tends to perform poorly in practice when compared to the reconstruction loss [2].
>
> > **Q1. Which representation is used for offline RL (i.e., policy learning)? I assume  $\hat s$ is used and not  $\hat e$, but I would like to clarify this.**
>
> Yes, $\hat s$ is used for offline RL. Please see L272 in the paper.
>
> > **Q2. The method requires extensive hyperparameter search over the variance and balancing factors of the KL divergence for each task, domain, and distraction type (Table 9). It would be much more convenient to eliminate this necessity. Do you have any ideas on how to achieve this?**
>
> Our method requires only three hyperparameters. In our experiments, we found that (1) the KL weights $c_E$ and $c_S$ are the most important and (2) the qualitative results (e.g. Figure 4) can be helpful in tuning hyperparameters efficiently.
>
> For the easy level, we can simply set $c_E = c_S = 1$ and only search over $\sigma^2 \in \{0.1, 0.02, 0.04 \}$.
> For the medium level, we searched only over $\sigma^2$ and $c_E $ while $c_S=1.0$ remains fixed. We found that without setting $c_E < c_S$, the model tends to put all representations inside $S$. Finally, for the hard level, we found that additional tuning of $c_S$ was necessary due to the fact that the size of the controllable agent now occupies a smaller segment of the image.
>
> We note that complete elimination of hyperparameter search is infeasible for both our method and other baselines since it is not possible to evaluate the policy offline.
>
> > **Q3. Since the method aims to improve representation learning, it could potentially be applied within an MBRL framework. How do you think it would perform when integrated with an MBRL framework?**
>
> We thank the reviewer for pointing out the potential applicability of our representation learning to (offline) MBRL framework. In principle, we can use the variational distribution $q_\phi(\hat s_t|\hat s_{t-1},a_{t-1})$ as the dynamics model for the MBRL framework and then train an additional reward predictor. While it can be leveraged, offline MBRL challenges still remain on the implementation side e.g. which pessimism heuristics to be used, the weight for the pessimism heuristics, as well as the percentage between the real trajectory and “imagined” trajectory.

---

> ### Author Response · Authors · 2024-11-22
>
> > **Q4. How can this method handle other forms of nuisance information, such as color perturbations of task-relevant parts, where the assumption in L278 does not hold?**
>
> In L278, we assume the state variables and exogenous variables occupy different parts of the visual observation thus we employ a compositional decoder commonly used in object-centric representation learning as an architectural choice which does not affect the main proposed objective.
>
> Following along with the example of color perturbations, the answer depends on how the color perturbs the visual observation. We characterize distraction as a factor that 1) does not affect the reward function, 2) is unaffected by action, 3) is independent of the agent state, and 4) is present in the observation. If the color perturbation only changes the color of the agent, then the color is dependent on the agent states which violates the definition of distraction that we have established in the first paragraph of section 2.1. Thus, the color can no longer be called a distraction rather as part of the agent state.
>
> As an example, consider the floor part of the visual observation (Figure 4). Although the floor is uncontrollable, the inferred state captures the floor as a controllable part. This is because how the floor moves depends on how the agent moves.
>
> We also make an additional example where the floor does not correlate with the agent state in the Finger environment (Appendix F in the updated manuscript). In such a case, the floor is inferred as part of the exogenous variables.
>
> > **Q5. Including more detailed descriptions of each column in the caption of Table 3 would make it much easier to interpret the results.**
>
> Thank you for the feedback. We have modified the caption of Table 3 in the updated manuscript.
>
> **References**
>
> [1] "Information prioritization through empowerment in visual model-based RL." Bharadhwaj et al. (ICLR 2022)
>
> [2] “Dream to control: Learning behaviors by latent imagination.” Hafner et al. (ICLR 2020)
>
> [3] “RePo: Resilient Model-Based Reinforcement Learning by Regularizing Posterior Predictability” Zhu et al. (NeurIPS 2023)

---

> > ### Comment · Reviewer_UCLi · 2024-11-27
> >
> > Thank you for your time and effort in addressing my concerns and questions.
> >
> > - A task in the new table should be listed as Cartpole, not Cheetah.
> >
> > - I appreciate your effort in conducting experiments with an additional baseline. However, I have concerns about the credibility of the results. InfoPower has been shown to perform well in the presence of video distractions in the background (Fig. 4 in the original paper), but the reproduced results suggest it is failing. While it is unfortunate that the original code is not publicly available, I worry that the current results may give readers the wrong impression. I recommend specifying in the main text that the results are based on a reimplementation.
> >
> > - The question about the prior work [2] has not been fully addressed. Could you provide a methodological comparison between this work and [2]? Since the code for [2] is publicly available, including it as a baseline would significantly strengthen the paper.
> >
> > The other questions were addressed quite well. I will maintain my current score.

---

> > > ### Author Response · Authors · 2024-11-29
> > >
> > > We thank the reviewer for actively giving us feedback during the discussion period.
> > >
> > > > **1. A task in the new table should be listed as Cartpole, not Cheetah.**
> > >
> > > Thank you for pointing out the mistake. We have fixed this directly on OpenReview.
> > >
> > > > **2. I appreciate your effort in conducting experiments with an additional baseline. However, I have concerns about the credibility of the results. InfoPower has been shown to perform well in the presence of video distractions in the background (Fig. 4 in the original paper), but the reproduced results suggest it is failing. While it is unfortunate that the original code is not publicly available, I worry that the current results may give readers the wrong impression. I recommend specifying in the main text that the results are based on a reimplementation.**
> > >
> > > Thank you for the suggestion. We agree with the reviewer’s recommendation to clarify this issue in the manuscript. We will specify that the results are based on our reimplementation, made to the best of our ability given the limited implementation details provided in the paper, as the source code is unavailable. We verified our implementation by ensuring the ground-truth state regression produced values within a reasonable range.
> > >
> > > Also, please note that both contrastive loss and reconstruction loss serve as lower bounds for mutual information. However, in practice, contrastive loss often underperforms compared to reconstruction loss [1]. Additionally, the analysis in Section 2.2 still holds where superfluous information remains within the learned representation. Regularizing a single latent variable with inverse dynamics alone is insufficient to induce minimal representation, as the representation can still predict actions given the state and next state. A method to eliminate this superfluous information is necessary, as we have done in our approach.

---

> > > > ### Author Response · Authors · 2024-11-29
> > > >
> > > > > **3. The question about the prior work [2] has not been fully addressed. Could you provide a methodological comparison between this work and [2]? Since the code for [2] is publicly available, including it as a baseline would significantly strengthen the paper.**
> > > >
> > > > **Experiment Result**
> > > >
> > > > | Task | | &nbsp;&nbsp;&nbsp;IFactor&nbsp;&nbsp;&nbsp; | &nbsp;&nbsp;&nbsp;CLEAR |
> > > > |-|:-:|:-:|:-:|
> > > > | Hopper | Clean (easy) | 32.4 ± 8.9 | 104.9 ± 2.8 |
> > > > | | SV (medium) | 17.3 $\pm$ 5.7 | 60.5 ± 5.1 |
> > > > | | MV (medium)| 2.4 $\pm$ 1.6 | 39.8 ± 11.5 |
> > > > | | 2 × 2 (hard) | 8.9 $\pm$ 2.0 | 50.5 ± 4.2 |
> > > > | Walker | Clean (easy) | 65.6 $\pm$ 2.1 | 89.9 ± 2.0 |
> > > > | | SV (medium) | 72.0 $\pm$ 1.8 | 87.6 ± 3.8 |
> > > > | | MV (medium) | 64.6 $\pm$ 2.9 | 88.4 ± 1.8 |
> > > > | | 2 × 2 (hard) | 29.3 $\pm$ 2.3 | 88.8 ± 2.4 |
> > > > | Cheetah | Clean (easy) | 54.0 $\pm$ 6.2 | 96.5 ± 0.6 |
> > > > | | SV (medium) | 58.3 $\pm$ 0.9 | 96.7 ± 1.5 |
> > > > | | MV (medium) | 45.6 $\pm$ 2.3 | 95.8 ± 1.1 |
> > > > | | 2 × 2 (hard) | 41.1 $\pm$ 2.9 | 79.1 ± 4.2 |
> > > >
> > > > The table shows the result of running IFactor [2] on the Hopper, Walker, and Cheetah tasks. We note that the result is similar to TiA [3] and DenoisedMDP [4] which intend to capture task-relevant representation.
> > > >
> > > > In this experiment, we use the hyperparameters given in the paper where $ \beta_1 = \beta_2 = \beta_3 = \beta_4 = 1 $ for the Clean; $ \beta_1 = \beta_2 = 1, \beta_3 = \beta_4 = 0.25 $ for the SV and MV; and $ \beta_1 = \beta_2 = 2, \beta_3 = \beta_4 = 0.25 $ for the 2x2. For all tasks and distraction types, we use $ \lambda_1 = \lambda_2 = 0 $. We refer the reviewer to [2] for further explanation of the hyperparameters.
> > > >
> > > > **Methodological Comparison**
> > > >
> > > > Similar to DenoisedMDP, IFactor categorizes state variables based on controllability and task relevance. Unlike DenoisedMDP, IFactor models four latent variables using this criterion. It applies four regularization terms to adjust the mutual information with reward (task relevance) or action (controllability).
> > > >
> > > > IFactor aims to learn representations that capture all task-relevant factors, regardless of controllability, similar to TiA and DenoisedMDP. In contrast, CLEAR focuses on capturing all non-distracting factors, where distractions are defined as factors that: (1) do not affect the reward function, (2) are unaffected by actions, (3) are independent of the agent’s state, and (4) are present in the observations. Both approaches use these representations as input for the RL agent.
> > > >
> > > > From a practical perspective, IFactor models four random variables, increasing the number of hyperparameters to tune—six in IFactor compared to three in CLEAR. The risk of "representation collapse", where information concentrates in a single latent variable, in models with multiple latent variables further complicates tuning. As a result, IFactor is significantly more challenging to tune due to the increased hyperparameter count.
> > > >
> > > > Furthermore, IFactor uses mutual information between certain latent variables and actions as regularization, making the model dependent on the data-generating policy. While this dependency may not be problematic in online or on-policy settings, it can be an issue in offline settings.
> > > >
> > > > **References**
> > > >
> > > > [1] “Dream to control: Learning behaviors by latent imagination.” Hafner et al. (ICLR 2020)
> > > >
> > > > [2] "Learning world models with identifiable factorization." Liu et al. (NeurIPS 2023)
> > > >
> > > > [3] “Learning task informed abstractions.” Fu et al. (ICML 2021)
> > > >
> > > > [4] “Denoised mdps: Learning world models better than the world itself.” Wang et al. (ICML 2022)

---

> > > > > ### Author Response · Authors · 2024-12-03
> > > > >
> > > > > With the deadline approaching in less than 8 hours, we would like to remind you that we are happy to promptly address any remaining concerns. Additionally, we kindly request that you take into account the additional baseline results we have provided [2].
> > > > >
> > > > > Thank you for your time and consideration.
> > > > >
> > > > > **References**
> > > > >
> > > > > [2] "Learning world models with identifiable factorization." Liu et al. (NeurIPS 2023)

---

### Official Review · Reviewer_7K7j · 2024-11-03

**Soundness:** 3
**Presentation:** 3
**Contribution:** 3
**Rating:** 6
**Confidence:** 2

**Summary:**

The paper aims at disentangling distractions in the visual observations from agent-related information for visual offline RL tasks. To this end, the paper formalizes the visual offline RL setting as an ExoPOMDP and derive the corresponding objective with the tools of information theory. Specifically, the disentanglement is achieved by a VAE-like model, where the decoder network has a compositional architecture that blends the reconstructed foreground and background visual parts with masks.

**Strengths:**

- The idea is simple and effective.
- The experiments seem thorough on the DeepMind Control Suite.
- The paper is clearly written and well-organized.

**Weaknesses:**

- I am admittedly not an expert of visual offline RL, and would like to hear my colleague reviewers' opinions on this.

**Questions:**

None

---

> ### Author Response · Authors · 2024-11-22
>
> # Response to Reviewer $\text{\textcolor{red}{7K7j}}$
> Thank you for taking the time to review our submission. Please let us know if there are further questions or necessary clarifications and we would be happy to address them during the discussion period.

---

### Author Response · Authors · 2024-11-22

# General Response

We would like to thank all reviewers for taking their time to provide constructive feedback on our paper. We are very encouraged that the reviewers have identified the following strengths in our work:

* Simple and effective idea with good motivation (Reviewer $\text{\textcolor{red}{7K7j}}$, $\text{\textcolor{blue}{UCLi}}$, $\text{\textcolor{green}{UDWw}}$)
* Extensive experiments on the DeepMind Control Suite (Reviewer $\text{\textcolor{red}{7K7j}}$, $\text{\textcolor{blue}{UCLi}}$)
* Comprehensive set of baselines and types of distractions (Reviewer $\text{\textcolor{blue}{UCLi}}$)
* Clearly written (Reviewer $\text{\textcolor{red}{7K7j}}$, $\text{\textcolor{green}{UDWw}}$)


We outline below the changes made to the manuscript:

1. Added InfoPower [1] as an additional baseline in the Related Work in Section 4 as well as the Experiments Section (Table 1 and 2 in Section 5) [$\text{\textcolor{blue}{UCLi}}$]

2. Added more detailed descriptions in the caption of Table 3 in Section 5.3 [$\text{\textcolor{blue}{UCLi}}$]

3. Added two additional environments: Finger-Spin and Cartpole-Swingup (Table 7 in Appendix F) [$\text{\textcolor{blue}{UCLi}}$, $\text{\textcolor{green}{UDWw}}$]

4. Added qualitative results for Finger-Spin (Figure 7 in Appendix F) [$\text{\textcolor{blue}{UCLi}}$, $\text{\textcolor{green}{UDWw}}$]

Finally, we have addressed each question as a separate response for each reviewer. Please let us know if there are any further questions or necessary clarifications and we would be happy to address them during the discussion period.

**References**

[1] "Information prioritization through empowerment in visual model-based RL." Bharadhwaj et al. (ICLR 2022)

---

### Meta-Review · Area_Chair_1rq7 · 2024-12-23

**Metareview:**

This paper looks at visual offline reinforcement learning under distractions in the visual observations. An ExoPOMDP (POMDP with exogenous variables) formulation is developed, leading to an information theoretic approach that disentangles distracting factors (that are not relevant for control) from agent-centric representations (that are). This leads to a dual-encoder architecture with a compositional decoder, and results are shown (initially) across three environments of DeepMind Control Suite, and subsequently add additional ones post-review.

The reviewers appreciated the problem setting, information-theoretic formulation, and subsequent approach that was developed. However, there are several concerns (some common) including limited environments (UCLi, UDWw), generality of the method and understanding exactly what kinds of visual perturbations can be handled (UCLi, UDWw), missing related work and benchmarks (UCLi), and hyper-parameters and their sensitivity/tuning. The authors provided a rebuttal, including some additional environments, qualitative results, and additional baselines. Reviewr UCLi mentioned concerns about these results, and during discussion UDWw also continued to have reservations.

  Overall, this paper is borderline and required considering all of the materials including the paper itself, reviews, rebuttals, author comments, and discussions. After considering these, I lean towards not accepting this paper at this time. While the paper's information-theoretic formulation and approach is interesting, the biggest overall limitation is the experimental execution. I agree with reviewers that the first three environments were indeed not sufficient to elucidate in a rigorous way when and how the method works compared to state-of-art baselines. The addition of new environments and baseline is appreciated, but there are significant concerns on the failure of InfoPower (original environments) and InfoGating (new environments). Unfortunately the rebuttal period doesn't lend itself to thorough examination of the correctness and cause of these results. Further, questions related to exactly what types of distractions the method supports should be better made explicit and analyzed throughout the paper. As a result, while we appreciate the author's significant work and rebuttal, the paper requires a more comprehensive empirical methodology and analysis to scientifically understand where and when this method works compared to state-of-art.

**Additional Comments On Reviewer Discussion:**

Reviewers raised a number of concerns, especially common ones regarding lack of a comprehensive suite of tasks to better understand when and where the method works and in comparison to state-of-art. Overall, this weighted the paper towards rejection at this time.

---

### Decision · Program_Chairs · 2025-01-22

Reject